# MoNE: Replacing Redundant Experts with Lightweight Novices for Structured Pruning of MoE

**Geng Zhang**[*], **Yuxuan Han**[*], **Yuxuan Lou, Yiqi Zhang, Wangbo Zhao**[†], **Yang You**[†]
National University of Singapore
{zhangg,youy}@comp.nus.edu.sg   wangbo.zhao96@gmail.com
{han_yuxuan,yuxuanlou,yiqi.zhang}@u.nus.edu

## Abstract

Mixture-of-Experts (MoE) enables efficient scaling of large language models by activating only a subset of experts per input token. However, deploying MoE-based models incurs significant memory overhead due to the need to retain all experts in memory. While structured pruning is promising to reduce memory costs, existing methods often show suboptimal performance and unstable degradation in three dimensions: model architectures, calibration data sources, and calibration sample sizes. This paper proposes **M**ixture-**o**f-**N**ovices-and-**E**xperts (**MoNE**), a novel expert pruning method that replaces redundant experts with lightweight novices to achieve effective and robust model compression. MoNE evaluates expert redundancy based on two metrics: access frequency and output variance. Experts exhibiting low usage and stable outputs are pruned and replaced with lightweight novices—unbiased estimations of their original outputs—minimizing performance degradation. Extensive experiments demonstrate that MoNE consistently outperforms baseline methods with minimal accuracy degradation across the three dimensions, confirming its effectiveness and robustness. Notably, it outperforms baselines by up to 2.72 for the average zero shot accuracy across nine downstream tasks under 25% pruning ratio, with only 0.14 performance drop for Qwen2-57B-A14B. The code is available at https://github.com/zxgx/mode-pd.

## 1 Introduction

Mixture-of-Experts (MoE) has emerged as a powerful architecture for advancing the capabilities of large language models (LLMs) (Liu et al., 2024; 2025; Muennighoff et al., 2025). MoE-based LLMs achieve higher parameter efficiency than vanilla transformer-based LLMs by replacing the MLP module with a set of smaller MLP modules (experts) and sparsely activating partial experts for each input token (Lepikhin et al., 2021). Despite its performance benefits, the deployment of MoE-based models often incur additional memory overhead to maintain the non-activated experts in memory, which is valuable but limited for existing accelerators such as GPU and TPU (Jouppi et al., 2023).

While diverse structured pruning methods have been proposed to reduce deployment memory costs by removing different model components while minimizing the performance degradation (Voita et al., 2019; He et al., 2024; Xia et al., 2024; Zhao et al., 2026; 2025a;b), we observe that these approaches often exhibit *suboptimal performance and unstable degradation* when applied to different MoE models. Specifically, we identify three critical dimensions where existing methods fall short: model architectures, calibration data sources and calibration sample sizes, as shown by experiments in Section 5.3. These limitations are evident across two main categories of structured pruning approaches for MoE models: general structured pruning and expert pruning as shown in Figure 1 (a). First, general structured pruning methods that remove model layers (Angular (Gromov et al., 2025)) or weight matrix channels (FLAP (An et al., 2024)) fail to account for the sparse computation scheme of MoE models when evaluating model component importance, resulting in inconsistent performance drop across the aforementioned three dimensions. Second, existing expert pruning

---

[*]Equal contribution. [†]Corresponding author.

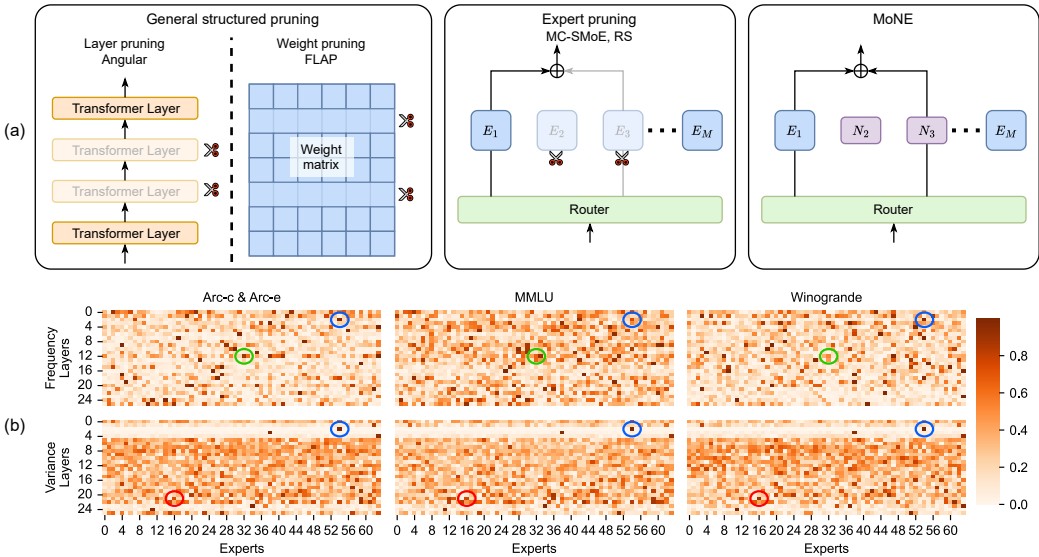

Figure 1: (a) Different structured pruning methods. (b) Layer-wise normalized expert access frequency and output variance of Deepseek-V2-Lite for three downstream tasks. Experts with high access frequency or output variances are the same across downstream tasks. Expert in blue circles has both high frequency and variance. Expert in red circles only has high variance. Expert in green circles only has high frequency. Similar observations on other models and tasks are in Appendix D.

methods such as MC-SMoE (Li et al., 2024) and RS (He et al., 2024) remove experts from MoE models primarily based on the expert access frequency. However, as shown in Figure 1 (b), this feature alone fails to fully capture the expert redundancy. Besides, these methods lack mechanisms to recover the performance loss caused by pruning.

To improve the effectiveness and robustness of structured pruning for MoE models, this paper proposes a novel expert pruning method, **M**ixture-**of**-**N**ovices-and-**E**xperts (**MoNE**) which replaces redundant experts with a lightweight structure, *novice*. Specifically, to prune an MoE model with MoNE, it first evaluates the expert redundancy by the access frequency and the output variance for each expert on a calibration dataset. Then, it identifies and prunes redundant experts that show low access frequency and stable output activations to reduce the memory overhead from redundant experts. Finally, the unbiased estimation of the pruned expert output is employed as the lightweight novice to reclaim the performance loss caused by the pruned expert. The intuition behind MoNE is that experts with low access frequency contribute less to the final outputs and experts whose outputs have low variance can be replaced with a constant but introduce less discrepancy. Moreover, Figure 1 (b) reveals that experts with less redundancy identified by MoNE exhibit strong consistency across various downstream tasks.

The contribution of this paper is summarized as follows:

- We propose a novel expert pruning method named MoNE which replaces redundant experts with lightweight novices to compress MoE models with minimal performance loss.

- We exploit the expert access frequency and output variance to measure the expert redundancy and employ the unbiased estimation of the expert output to minimize the output discrepancy after pruning, thus achieving effective and robust pruning results.

- Extensive experiment results demonstrate that MoNE consistently outperforms baseline methods under varying MoE architectures, calibration data sources and calibration sample sizes. Notably, it outperforms baselines by up to 2.72 for the average zero shot accuracy across nine downstream tasks under 25% pruning ratio, with only 0.14 performance drop for Qwen2-57B-A14B.

## 2 RELATED WORK

Model pruning compresses a model by removing certain redundant model parameters while preserving accuracy. Existing pruning methods generally fall into two categories: *unstructured pruning* and *structured pruning*. Unstructured pruning eliminates any model parameter that has minimal impact on model performance. Methods such as SparseGPT (Frantar & Alistarh, 2023), Wanda (Sun et al., 2024), SparseLLM (Bai et al., 2024) excel in maintaining accuracy while achieving high compression ratios. However, the resulting irregular sparsity patterns hinder efficient representation and execution on hardware accelerators.

In contrast, structured pruning removes certain modules of a model, preserving hardware-friendly structures. Early researches prune redundant transformer layers of a LLM (Fan et al., 2020; Ling et al., 2024; Gromov et al., 2025). LLM-Pruner (Ma et al., 2023), FLAP (An et al., 2024), MoE-Pruner (Xie et al., 2024) and SlimMoE (Li et al., 2025) remove rows or columns of individual weight matrices. Recent work also proposes to delete components such as attention, MLP or MoE modules within each transformer layer (Voita et al., 2019; He et al., 2024). Minitron (Muralidharan et al., 2024) and Sheared LLaMA (Xia et al., 2024) combine different granularity and automatically search for the optimal structures to prune. Despite their versatility, existing structured pruning methods often exhibit inconsistent performance across MoE architectures.

This work focuses on expert pruning, a unique direction of structured pruning for MoE models (Lepikhin et al., 2021; Liu et al., 2024). Expert pruning targets on deleting individual experts for each layer to compress an MoE model. Previous expert pruning methods either require exhaustive search to identify redundant experts (Lu et al., 2024), or heavily rely on retraining to recover accuracy due to the suboptimal pruning performance (Li et al., 2024; He et al., 2024). However, the exhaustive search is not applicable to modern MoE model architectures such as Deepseek (DeepSeek-AI, 2024; Liu et al., 2025), OLMoE (Muennighoff et al., 2025) or Qwen (Qwen, 2024; Yang et al., 2025), as their MoE layer contains 64 experts or even more, yielding a tremendous search space that is intractable. Retraining obscures the advantages of expert pruning over other structured pruning methods.

## 3 PRELIMINARIES

### 3.1 MIXTURE-OF-EXPERTS (MOE)

MoE-based LLMs replace the traditional MLP module in the transformer layer with MoE module. Each MoE module consists of a router network $G$ and a set of experts $\mathcal{E} = \{E_1, E_2, \ldots, E_M\}$, where $M$ is the number of experts and each expert is a smaller MLP. Let $\mathbf{x} \in \mathbb{R}^d$ be the hidden state of an input token, where $d$ is the hidden size of the model, the output of an MoE module is computed as:

$$\text{MoE}(\mathbf{x}, G, \mathcal{E}) = \sum_{E_i \in \mathcal{S}_{k,\mathbf{x}}} G_i(\mathbf{x}) \cdot E_i(\mathbf{x}) \qquad (1)$$

The output of the router network $G(\mathbf{x}) \in \mathbb{R}^M$ represents the routing scores for all experts, and $\mathcal{S}_{k,\mathbf{x}} \subseteq \mathcal{E}$ denotes the top $k$ experts with the highest routing scores for input $\mathbf{x}$. The final output of the MoE module is the weighted sum of outputs from the top $k$ experts. While Equation 1 captures the general MoE computation, implementations for $G$ and $E_i$ may vary across model architectures (DeepSeek-AI, 2024; Liu et al., 2024; Muennighoff et al., 2025; Liu et al., 2025).

### 3.2 EXPERT PRUNING FORMULATION

Previous studies have revealed that not all experts contribute equally, and pruning less important ones can reduce memory overhead with marginal performance degradation (Lu et al., 2024; Li et al., 2024; Huang et al., 2025). However, searching for the target experts to prune at the global model perspective falls into a tremendous search space, as the number of experts per transformer layer increases with the evolving of the MoE model architectures (Lepikhin et al., 2021; Jiang et al., 2024; DeepSeek-AI, 2024; Liu et al., 2024). Following the layer-wise pruning scheme (Frantar & Alistarh, 2023; An et al., 2024; Lu et al., 2024; Ling et al., 2024), our goal of expert pruning is to identify a subset of redundant experts $\mathcal{P} \subseteq \mathcal{E}$ such that we can minimize the output difference after compressing their parameters:

$$\min_{\mathcal{P} \subseteq \mathcal{E}} \|\text{MoE}(\mathbf{x}, G, \mathcal{E} \setminus \mathcal{P}) - \text{MoE}(\mathbf{x}, G, \mathcal{E})\|_2 \qquad (2)$$

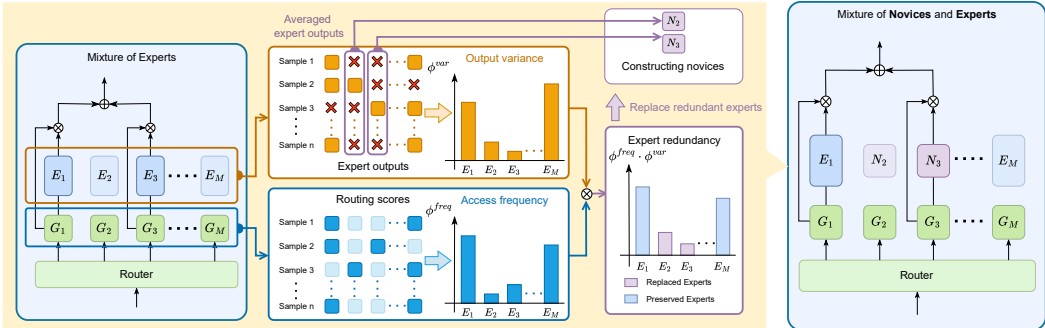

Figure 2: **The overview of MoNE.** Given an MoE model, it first exploits a calibration dataset to evaluate the expert access frequency and output variance. Then, the two metrics are fused to get the expert redundancy. Finally, the novices are derived from the averaged outputs for redundant experts.

To achieve this goal, the core problem is twofold: (1) find *a metric to evaluate the importance of the experts in each layer*, so that we can identify the expert subset $\mathcal{P}$, and (2) find *an pruning method to compress the parameters of $\mathcal{P}$*, so that we can reduce the model size. While $\mathcal{E} \setminus \mathcal{P}$ implies directly removing redundant experts (He et al., 2024; Lu et al., 2024), existing methods have also explored expert merging to mitigate the expert redundancy (Li et al., 2024).

## 4 MIXTURE OF NOVICES AND EXPERTS

This section introduces **Mixture-of-Novices-and-Experts (MoNE)**, a novel expert pruning method designed to achieve effective and robust compression for MoE models while minimizing performance degradation. Section 4.1 presents the computational framework of MoNE. Section 4.2 defines the metric to evaluate the redundancy of experts. Section 4.3 explains the pruning process that compresses the redundant experts to lightweight novices. The overview of MoNE is depicted in Figure 2.

### 4.1 MONE FRAMEWORK

MoE models are often trained with auxiliary losses to ensure load balance among experts in each layer, enabling each expert to learn certain aspect of knowledge (Lepikhin et al., 2021; DeepSeek-AI, 2024; Muennighoff et al., 2025). However, most existing expert pruning methods directly remove experts (He et al., 2024; Li et al., 2024), often leading to inconsistent performance drops across different model architectures or calibration data. MoNE addresses this issue by introducing lightweight structures called **novices** to replace the pruned experts. A novice is designed to capture the essential knowledge previously held by the removed experts. In contrast to simply removing redundant experts, MoNE compensates for knowledge loss by leveraging novices, thereby preserving the overall performance of the model while maintaining compression efficiency. Specifically, the output of the MoNE is computed as:

$$\text{MoNE}(\mathbf{x}) = \left( \sum_{E_i \in \mathcal{S}_{k,\mathbf{x}} \setminus \mathcal{P}} G_i(\mathbf{x}) \cdot E_i(\mathbf{x}) \right) + \left( \sum_{E_i \in \mathcal{S}_{k,\mathbf{x}} \cap \mathcal{P}} G_i(\mathbf{x}) \cdot N_i \right) \tag{3}$$

where $\mathcal{S}_{k,\mathbf{x}} \setminus \mathcal{P}$ and $\mathcal{S}_{k,\mathbf{x}} \cap \mathcal{P}$ denote the preserved and pruned experts among the top $k$ activated experts respectively. $N_i \in \mathbb{R}^d$ is the novice $i$ that retains the essential knowledge of the pruned expert. Notably, $N_i$ is a compressed vector that does not involve any computation with the input token $\mathbf{x}$. As a result, the computation and memory overhead is nearly identical to directly removing experts. Furthermore, replacing experts with novices introduces adaptive computation overhead for different tokens, leading to fewer activated parameters for tokens routed to novices. Nevertheless, empirical results in Section 5.2 demonstrate that MoE models pruned by MoNE maintain more zero shot performance on downstream tasks compared to existing expert pruning methods that only remove experts but keep the same activated parameters.

### 4.2 Expert Redundancy Evaluation

To identify the expert subset $\mathcal{P}$, we introduce an *expert redundancy score* $\phi$ to assess the redundancy of experts. To ensure the pruned experts contribute minimally to the model's overall performance, the expert redundancy score $\phi$ takes two aspects into consideration: the variance in an expert's output across a calibration dataset $\mathcal{C}$, and the frequency of an expert selected by the router network $G$.

**Variance-based redundancy** As the novices are constant vectors to ensure reduced computation and memory overhead, the outputs of the pruned experts are expected to have low variance across a calibration dataset $\mathcal{C}$. In other words, experts with high output variance should be retained to contribute more discriminative information during inference, whereas experts with low output variance could be compressed into a more efficient representation, i.e., a novice. The second row of Figure 1 (b) visualizes this motivation. Expert outputs exhibit diverse variances, but we can find experts in blue and red circles that *maintain high variances across different downstream tasks*. Therefore, we introduce a variance-based redundancy $\phi_i^{var}$ to measure the output variance for expert $E_i$. Concretely, $\phi_i^{var}$ is the L2 norm of the unbiased estimation for the output variance:

$$\phi_i^{var} = \left\| \sqrt{\frac{\sum_{\mathbf{x} \in \mathcal{C}} (E_i(\mathbf{x}) - \overline{E_i})^2 \cdot \mathbb{I}(E_i \in \mathcal{S}_{k,\mathbf{x}})}{\sum_{\mathbf{x} \in \mathcal{C}} \mathbb{I}(E_i \in \mathcal{S}_{k,\mathbf{x}}) - 1}} \right\|_2 \tag{4}$$

$$\overline{E_i} = \frac{\sum_{\mathbf{x} \in \mathcal{C}} E_i(\mathbf{x}) \cdot \mathbb{I}(E_i \in \mathcal{S}_{k,\mathbf{x}})}{\sum_{\mathbf{x} \in \mathcal{C}} \mathbb{I}(E_i \in \mathcal{S}_{k,\mathbf{x}})} \tag{5}$$

where $\mathbb{I}(E_i \in \mathcal{S}_{k,\mathbf{x}})$ is the indicator function to show whether $E_i$ is among top $k$ experts for the input token $\mathbf{x}$ of the calibration dataset $\mathcal{C}$.

**Frequency-based redundancy** The routing scores and access frequencies of the router network $G$ serve as strong indicators of the overall redundancy of an expert (He et al., 2024; Li et al., 2024). Intuitively, experts which are rarely selected or consistently assigned lower routing scores are likely to have a minimal impact on the model's output. As shown in Figure 1 (b), we can identify typical experts in blue and green circles that show consistent high frequency over the three downstream tasks. Notably, the expert in green circles only has high frequency. Therefore, *the frequency and variance information can complement the discrepancy ignored by each other.* Based on this observation, we define the frequency-based redundancy $\phi_i^{freq}$ of the expert $E_i$ as the average routing score across a calibration dataset $\mathcal{C}$ of which $E_i$ is among the top $k$ selected experts. Formally, the frequency-based redundancy $\phi_i^{freq}$ is defined as:

$$\phi_i^{freq} = \frac{\sum_{\mathbf{x} \in \mathcal{C}} G_i(\mathbf{x}) \cdot \mathbb{I}(E_i \in \mathcal{S}_{k,\mathbf{x}})}{\sum_{\mathbf{x} \in \mathcal{C}} \mathbb{I}(E_i \in \mathcal{S}_{k,\mathbf{x}})} \tag{6}$$

Finally, the two redundancy metrics are fused to obtain the expert redundancy score $\phi$:

$$\phi = \phi^{var} \cdot \phi^{freq} \tag{7}$$

A lower expert redundancy score $\phi_i$ indicates higher redundancy for expert $E_i$, making it a suitable candidate for pruning and replacement with a novice $N_i$.

### 4.3 Expert Replacement with Novice

After identifying the pruned expert subset $\mathcal{P}$, we need to construct lightweight novices to replace them. According to Equation 2, the general objective for expert pruning is to minimize the discrepancy introduced by the removed expert outputs. Since the output after applying MoNE is formulated as Equation 3, the concrete objective for MoNE can be translated to:

$$\min_{E_i \in \mathcal{P}} \sum_{\mathbf{x} \in \mathcal{C}} (\|E_i(\mathbf{x}) - N_i\|_2) \tag{8}$$

Because $N_i$ is a constant vector, the optimal novice vector $N_i$ that best approximates the output of a pruned expert $E_i$ can be obtained in a closed form, i.e., $\overline{E_i}$ in Equation 5.

To sum up, MoNE uses the unbiased estimations of mean expert outputs to replace experts that have the minimum output variance. As a result, MoNE achieve the goal that effectively and robustly compresses the MoE experts while minimizing performance degradation.

## 5 EVALUATION

### 5.1 EXPERIMENT SETUP

**Base MoE models**    To validate the effectiveness and robustness of MoNE, we conducted structured pruning on five open source MoE models with diverse architectures and model scales: **OLMoE** (Muennighoff et al., 2025), **Moonlight** (Liu et al., 2025), **DeepSeek-V2-Lite** (DeepSeek-AI, 2024), **Qwen2-57B-A14B** (Qwen, 2024) and **Qwen3-30B-A3B** (Yang et al., 2025). OLMoE has 7B parameters with 1B activated parameters per token. Both Moonlight and Deepseek-V2-Lite have 16B parameters with 3B activated pamaters per token. OLMoE and Moonlight represent SOTA MoE models at their respective scales. To demonstrate scalability to larger architectures, we also consider the Qwen series: Qwen3-30B-A3B with 30B parameters and 3B activated per token, and Qwen2-57B-A14B with 57B parameters and 14B activated per token. We chose the base version of the five models for experiments.

**Baseline methods**    We selected structured pruning methods for different structures as baseline. Notably, unless explicitly stated, we did not apply any weight update to compare the effect of pruning methods. Specifically, for general structured pruning methods, we used **Angular** for layer pruning (Gromov et al., 2025), which evaluates the layer importance by the angular distance between the input activations for different layers, and we used **FLAP** for weight pruning (An et al., 2024), which evaluates the channel importance by the fluctuation of the input activations and compensates the performance loss with the averaged output activations. For expert pruning methods, we adopted the expert merging method in **MC-SMoE** (Li et al., 2024) for one of the expert pruning baselines. Another expert pruning baseline is **RS** (He et al., 2024), which uses routing scores to evaluate the expert importance and discards less accessed ones.

**Implementation details**    We tested two pruning ratios: 25% and 50%. To demonstrate the robustness of MoNE to calibration data, we conducted experiments on two calibration data sources: Zyda2 (Tokpanov et al., 2024) and C4 (Raffel et al., 2020). Both datasets are constructed for LLM pretraining and C4 is commonly used as the calibration dataset for model compression (Ling et al., 2024; Frantar & Alistarh, 2023; Gromov et al., 2025; Xia et al., 2024). Besides, we also investigated the performance under three calibration sample sizes: 100, 500 and 1000 in Section 5.3.

**Evaluation protocol**    Following previous researches (Ma et al., 2023; Bai et al., 2024; Ling et al., 2024; Xia et al., 2024; An et al., 2024), we adopted lm-evaluation-harness[1] (Gao et al., 2024) to measure the zero shot accuracy and average results on nine downstream tasks: Arc-c and Arc-e (Clark et al., 2018), BoolQ (Clark et al., 2019), COPA (Roemmele et al., 2011), MMLU (Hendrycks et al., 2021), OBQA (Mihaylov et al., 2018), PIQA (Bisk et al., 2020), RTE (Wang et al., 2019) and Winogrande (Sakaguchi et al., 2021).

Though more complex downstream tasks such as coding (Zhang et al., 2025), math (Lightman et al., 2023) or reasoning (Lin et al., 2025) exist, these tasks are still challenging for full LLMs (Yang et al., 2025; Liu et al., 2024). Moreover, existing study shows that model compression for complex tasks often requires additional task-specific fine-tuning (Sarkar et al., 2024; Chen et al., 2022), which is beyond the scope of this work and we consider it a promising direction of future work.

### 5.2 EFFECTIVENESS EVALUATION

This section validates the effectiveness of MoNE by comparing the zero shot performance of 25% pruned models with 100 calibration samples from the Zyda2 dataset. The results are presented in Table 1 and Table 2. Results under 50% pruning ratio are extended to Table 10 in Appendix C.

Table 1 indicates that **MoNE consistently outperforms baseline methods** in terms of the average accuracy on the nine tasks. In particular, it shows average accuracy improvement as large as **2.72** for the pruned OLMoE compared to baseline methods, and it incurs accuracy drop as small as only **0.14** for the pruned Qwen2-57B-A14B. Furthermore, MoNE-pruned models can achieve **either the best or the second best result for individual tasks under most settings**.

---

[1]https://github.com/EleutherAI/lm-evaluation-harness

Table 1: Zero shot performance with 100 calibration samples from Zyda2 dataset. Best results are in **bold**, and the second best are underlined. Green cells indicate results no less than original models.

(a) OLMoE

| Pruning ratio | Model/Method | Arc-c | Arc-e | BoolQ | COPA | MMLU | OBQA | PIQA | RTE | Winogrande | Avg. |
|---|---|---|---|---|---|---|---|---|---|---|---|
| 0% | OLMoE | 49.23 | 76.89 | 70.09 | 85.00 | 53.54 | 44.40 | 79.76 | 71.84 | 68.90 | 66.63 |
| 25% | Angular | 32.76 | 61.91 | 61.71 | 74.00 | 23.13 | 37.60 | 71.65 | 53.07 | 55.09 | 52.33 |
| | FLAP | 40.53 | **67.55** | 62.69 | 78.00 | **41.16** | 37.80 | 74.81 | 61.37 | 60.93 | 58.32 |
| | MC-SMoE | 35.67 | 54.92 | 63.49 | 73.00 | 29.04 | 30.60 | 67.19 | 55.23 | 65.75 | 52.77 |
| | RS | 25.85 | 43.01 | 59.08 | 74.00 | 29.63 | 36.20 | 66.16 | 56.68 | 59.98 | 50.07 |
| | MoNE (Ours) | **42.32** | 64.81 | **67.19** | 85.00 | 40.13 | **40.80** | **78.07** | **64.62** | **66.46** | **61.04** |

(b) Moonlight

| Pruning ratio | Model/Method | Arc-c | Arc-e | BoolQ | COPA | MMLU | OBQA | PIQA | RTE | Winogrande | Avg. |
|---|---|---|---|---|---|---|---|---|---|---|---|
| 0% | Moonlight | 58.28 | 82.49 | 80.40 | 92.00 | 67.30 | 45.60 | 81.12 | 65.70 | 71.11 | 71.56 |
| 25% | Angular | 39.76 | 52.69 | 38.90 | 79.00 | 42.57 | 32.20 | 68.50 | 61.01 | 62.04 | 52.96 |
| | FLAP | 48.55 | 76.01 | 75.93 | 90.00 | **55.84** | 42.20 | 77.97 | 64.26 | 68.19 | 66.55 |
| | MC-SMoE | 47.61 | 73.15 | 78.72 | 89.00 | 46.11 | 43.60 | 80.36 | 56.32 | 71.43 | 65.14 |
| | RS | 55.80 | **80.64** | 78.69 | 90.00 | 46.73 | 46.40 | **81.01** | 58.84 | **72.30** | 67.82 |
| | MoNE (Ours) | **55.89** | 80.60 | **79.57** | 90.00 | 55.23 | **46.80** | 80.85 | 61.01 | 71.98 | **69.10** |

(c) Deepseek-V2-Lite

| Pruning ratio | Model/Method | Arc-c | Arc-e | BoolQ | COPA | MMLU | OBQA | PIQA | RTE | Winogrande | Avg. |
|---|---|---|---|---|---|---|---|---|---|---|---|
| 0% | Deepseek-V2-Lite | 48.72 | 76.18 | 79.88 | 88.00 | 54.96 | 43.60 | 80.25 | 61.37 | 71.51 | 67.16 |
| 25% | Angular | 32.00 | 53.28 | 64.92 | 75.00 | 26.95 | 34.00 | 71.33 | 58.84 | 61.01 | 53.04 |
| | FLAP | 43.69 | 71.46 | 75.26 | 84.00 | 47.28 | 41.40 | 78.18 | **62.82** | 67.72 | 63.53 |
| | MC-SMoE | 36.69 | 60.77 | 71.31 | 84.00 | 42.22 | 36.60 | 75.57 | 58.48 | 68.67 | 59.37 |
| | RS | **49.32** | 74.41 | 69.39 | 90.00 | **50.35** | **43.80** | **80.14** | 62.09 | 70.24 | 65.53 |
| | MoNE (Ours) | 46.67 | **74.62** | **78.47** | 90.00 | 49.05 | 43.00 | 79.76 | 62.09 | 71.43 | **66.12** |

(d) Qwen2-57B-A14B

| Pruning ratio | Model/Method | Arc-c | Arc-e | BoolQ | COPA | MMLU | OBQA | PIQA | RTE | Winogrande | Avg. |
|---|---|---|---|---|---|---|---|---|---|---|---|
| 0% | Qwen2-57B-A14B | 49.66 | 69.44 | 86.45 | 93.00 | 74.06 | 44.20 | 81.23 | 74.73 | 74.27 | 71.89 |
| 25% | Angular | 29.44 | 54.17 | 59.51 | 70.00 | 23.92 | 32.80 | 70.02 | 54.87 | 49.57 | 49.37 |
| | FLAP | 50.00 | **72.85** | **86.91** | 91.00 | 65.02 | **45.40** | 81.12 | **77.62** | **76.09** | **71.78** |
| | MC-SMoE | 46.67 | 66.25 | 86.45 | 88.00 | 69.46 | 43.40 | 80.20 | 74.73 | 75.14 | 70.03 |
| | RS | 49.15 | 69.78 | 84.77 | 87.00 | 70.99 | 44.80 | **81.34** | 74.37 | 74.19 | 70.71 |
| | MoNE (Ours) | 49.66 | 68.73 | 86.88 | **94.00** | 71.64 | 45.20 | 81.07 | 75.45 | 73.09 | 71.75 |

(e) Qwen3-30B-A3B

| Pruning ratio | Model/Method | Arc-c | Arc-e | BoolQ | COPA | MMLU | OBQA | PIQA | RTE | Winogrande | Avg. |
|---|---|---|---|---|---|---|---|---|---|---|---|
| 0% | Qwen3-30B-A3B | 55.89 | 79.42 | 88.69 | 84.00 | 77.82 | 44.80 | 80.30 | 82.31 | 70.64 | 73.76 |
| 25% | Angular | 44.62 | 68.60 | 80.52 | 77.00 | 59.75 | 40.40 | 75.30 | 70.40 | 62.51 | 64.34 |
| | FLAP | 50.85 | 76.68 | 85.72 | 85.00 | 69.43 | 42.80 | 77.31 | **81.95** | 70.17 | 71.10 |
| | MC-SMoE | 52.73 | 76.98 | 88.75 | 83.00 | 72.25 | **44.40** | 79.71 | 80.87 | 70.40 | 72.12 |
| | RS | 53.75 | 78.32 | 88.53 | 85.00 | **74.60** | 43.00 | 79.92 | 79.78 | 69.61 | 72.50 |
| | MoNE (Ours) | **56.14** | 79.17 | **89.11** | 85.00 | 74.04 | 43.00 | 79.27 | 77.98 | **70.48** | **72.69** |

Table 2: Maximum pruning ratios with 1% accuracy loss after MoNE pruning using 100 calibration samples from Zyda-2 dataset.

| Model | Max. pruning ratio | Avg. perf before pruning | Avg. perf after pruning |
|---|---|---|---|
| OLMoE | 16% | 66.63 | 66.00 |
| Moonlight | 16% | 71.56 | 70.59 |
| Deepseek-V2-Lite | 20% | 67.16 | 66.31 |
| Qwen2-57B-A14B | 25% | 71.89 | 71.75 |
| Qwen3-30B-A3B | 24% | 73.76 | 73.61 |

An interesting observation is that all the three expert pruning methods, MC-SMoE, RS and MoNE can achieve results on par with or even better than the original models on certain tasks. The specific examples are shown in green background in Table 1. All these results indicate that there is indeed redundancy existing in the expert level for the examined MoE models, and expert pruning can rule

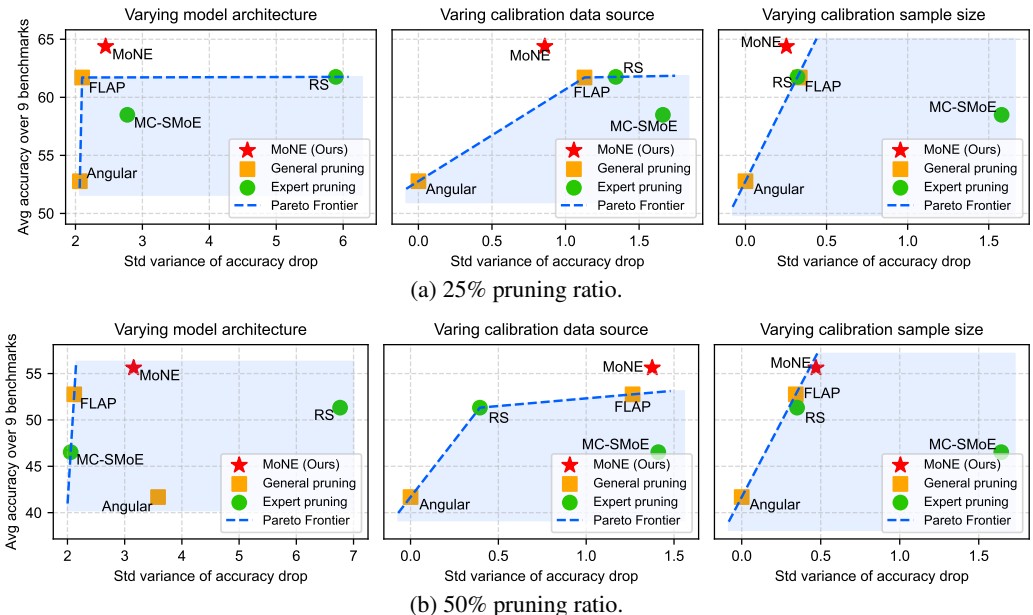

(a) 25% pruning ratio.

(b) 50% pruning ratio.

Figure 3: Average accuracy versus accuracy drop variance. MoNE advances the Pareto frontier across varying model architectures, calibration data sources and calibration sample sizes.

out such redundancy to achieve even better results on these tasks. Among the three expert pruning methods, MoNE consistently surpasses the two baseline methods, demonstrating its strong capability.

Table 2 evaluates the maximum pruning ratios that maintain accuracy loss within 1% across different MoE models. As shown in Table 2, larger and more powerful models such as Qwen2-57B-A14B and Qwen3-30B-A3B tolerate more aggressive pruning (25% and 24%, respectively) compared to smaller models like OLMoE, Moonlight, and Deepseek-V2-Lite (16–20%). This finding highlights the scalability of proposed MoNE method for large model scales. In addition, combining the observations from Table 1 and Table 2, we argue that larger MoE models may have stronger capability in language modeling but also contain increasing parameter redundancy at the expert level, and MoNE can efficiently eliminates such redundancy at minimal performance degradation.

### 5.3    ROBUSTNESS EVALUATION

This section evaluates the robustness of MoNE across three key dimensions: model architecture, calibration data source, and calibration sample size. Due to prohibitive compute for exhaustive experiments on large models, we tested three models: OLMoE, Moonlight and Deepseek-V2-Lite using 100, 500 and 1000 calibration samples on C4 and Zyda-2 dataset separately. For each dimension, we vary one factor while averaging results over the other two, measuring both average accuracy and the standard deviation of accuracy drop. The results are visualized in Figure 3, with detailed scores provided in Appendix C. As shown in Figure 3a, **MoNE advances the Pareto frontier across all three dimensions at the 25% pruning ratio**, demonstrating superior robustness and effectiveness compared to existing structured pruning methods. At the 50% pruning ratio (Figure 3b), MoNE exhibits slightly higher variance under varying model architectures and calibration sample sizes. Nevertheless, it remains the most effective method, **outperforming baseline methods by a significant margin of 2.85**.

### 5.4    ABLATION STUDY

This section presents the ablation study to evaluate the effects of the two redundancy metrics and the impact of novice replacement across the downstream tasks. Figure 4 displays the average accuracy drop relative to our proposed methods, with lower values indicating greater degradation. Results are averaged over the three evaluation dimensions to provide a robust assessment. We observe that integrating the fused expert redundancy score with novice replacement yields better performance, particularly under higher pruning ratios. This indicates that our approach is especially effective in

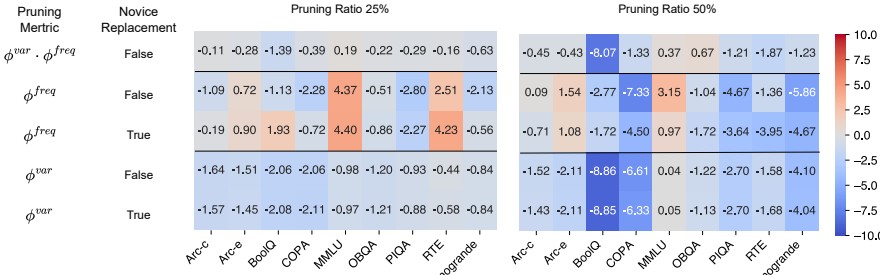

Figure 4: Ablation study on expert access frequency, output variance and novice replacement. Numbers are the difference to the proposed MoNE. The detailed result is provided in Appendix E.

Table 3: Zero shot performance of the 25% pruned OLMoE after continued pretraining with 2B tokens from OLMoE-mix-0924. Best results are in **bold**, and the second best are underlined.

| Model/Method | Arc-c | Arc-e | BoolQ | COPA | MMLU | OBQA | PIQA | RTE | Winogrande | Average |
|---|---|---|---|---|---|---|---|---|---|---|
| OLMoE | 49.23 | 76.89 | 70.09 | 85.00 | 53.54 | 44.40 | 79.76 | 71.84 | 68.90 | 66.63 |
| Angular | 38.82 | 64.69 | 63.52 | 82.00 | 25.42 | 39.80 | 76.50 | 51.62 | 59.04 | 55.71 |
| FLAP | 42.24 | 69.07 | 69.51 | 80.00 | **45.56** | 40.40 | 77.42 | 50.18 | 63.54 | 59.77 |
| MC-SMoE | 42.75 | 70.41 | 69.76 | 80.00 | 44.13 | 37.60 | 75.79 | 66.43 | 64.96 | 61.31 |
| RS | 44.97 | 72.94 | 70.73 | 85.00 | 43.28 | **43.00** | 78.67 | **72.20** | 65.98 | 64.09 |
| MoNE (Ours) | **47.35** | **74.33** | **71.56** | **87.00** | 43.30 | 40.40 | **78.89** | 67.51 | **67.25** | **64.18** |

preserving model quality when pruning is more aggressive. Notably, for tasks such as BoolQ, COPA, and PIQA, our proposed method outperforms the ablation baselines by a large margin—achieving accuracy gains of up to 8.85. However, for MMLU, pruning based solely on frequency appears to offer a slight advantage, suggesting that frequently activated experts may play a more critical role in domain-specific reasoning tasks.

## 5.5 ACCURACY RECOVERY WITH CONTINUED PRETRAINING

To evaluate performance recovery capabilities, we conducted continued pretraining on the 25% compressed OLMoE model pruned by 100 Zyda2 samples, as only this model releases its pretraining dataset, OLMoE-mix-0924[2]. The sequence length was set to 4096 and the global token size per step was 4M. Each pruned model was trained with 2B tokens, i.e., 512 steps, and the peak and minimum learning rate (lr) were 5e-5 and 5e-6, respectively. We employed the cosine lr scheduler with 50 warm up steps. Other hyperparameters were the same as the original configuration for OLMoE (Muennighoff et al., 2025). All the experiments could run on a single H20 GPU, but we accelerated the training with 16 H20 GPUs.

The results are summarized in Table 3. This table shows that MoNE achieves the average accuracy closest to the original model with only 2B tokens from a pretraining dataset, demonstrating the promising capability of the MoNE computation framework. Besides, MC-SMoE and RS reclaim 8.54 and 14.02 average accuracy, indicating that expert pruning is not only effective to eliminate redundancy, but also relatively easier to recover performance with continued pretraining.

## 6 CONCLUSION

In this paper, we propose MoNE, a novel expert pruning method that replaces redundant experts with lightweight novices to compress MoE models. MoNE evaluates expert redundancy based on expert access frequency and output variance in each model layer, pruning experts with low usage and stable outputs while replacing them with novices that provide unbiased output estimates. Extensive experiments across different MoE architectures, calibration data sources, and sample sizes demonstrate that MoNE outperforms existing structured pruning methods by maintaining higher zero-shot performance across nine downstream tasks.

---

[2] https://huggingface.co/datasets/allenai/OLMoE-mix-0924

ACKNOWLEDGMENT

We would like to acknowledge that computational work involved in this research work is supported by NUS IT's Research Computing group using grant numbers CFP02-CF-004. Yang You's research group is being sponsored by NUS startup grant (Presidential Young Professorship), Singapore MOE Tier-1 grant, ByteDance grant, NUS ARTIC grant, Apple grant, Alibaba grant, Google Research and Google grant for TPU usage.

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

## THE USE OF LARGE LANGUAGE MODELS

All writing, visualizations, and experiments are **completed by the authors**. LLMs (e.g., Claude) are used **solely to refine the writing**.

We organize our appendix as follows:

- Section A shows the evaluation of MoNE on domain-specific tasks.
- Section B presents ablation study on the pruning metric.
- Section C provides detailed scores for various models, calibration datasets and calibration sample sizes.
- Section D visualizes expert access frequency and output variance for more tasks and models.
- Section E presents the detailed scores for ablation study.
- Section F shows the inference latency and memory overhead benefits for the proposed MoNE.

Table 4: Base and Instruct OLMoE models on Math and GSM8K.

| Model | GSM8K | Math |
|---|---|---|
| *No calibration* | | |
| OLMoE | 52.77 | 15.78 |
| OLMoE-Instruct | 67.63 | 18.64 |
| *Calibration: Zyda2* | | |
| OLMoE | 3.34 | 1.94 |
| OLMoE-Instruct | 7.96 | 2.54 |
| *Calibration: Task-specific* | | |
| OLMoE-Instruct (Math) | 65.28 | 18.22 |
| OLMoE-Instruct (GSM8K) | 67.48 | 17.64 |

Table 5: Base and Instruct Moonlight models on Math and GSM8K.

| Model | GSM8K | Math |
|---|---|---|
| *No calibration* | | |
| Moonlight | 74.22 | 42.32 |
| Moonlight-Instruct | 77.03 | 39.26 |
| *Calibration: Zyda2* | | |
| Moonlight | 46.40 | 3.32 |
| Moonlight-Instruct | 51.86 | 6.64 |
| *Calibration: Task-specific* | | |
| Moonlight-Instruct (Math) | 75.28 | 38.56 |
| Moonlight-Instruct (GSM8K) | 76.72 | 35.84 |

Table 6: Base and Instruct DeepSeek-V2-Lite models on Math and GSM8K.

| Model | GSM8K | Math |
|---|---|---|
| *No calibration* | | |
| DeepSeek-V2-Lite | 38.82 | 16.54 |
| DeepSeek-V2-Lite-Chat | 66.49 | 18.56 |
| *Calibration: Zyda2* | | |
| DeepSeek-V2-Lite | 25.40 | 6.80 |
| DeepSeek-V2-Lite-Chat | 37.07 | 4.76 |
| *Calibration: Task-specific* | | |
| DeepSeek-V2-Lite-Chat (Math) | 64.44 | 20.76 |
| DeepSeek-V2-Lite-Chat (GSM8K) | 63.76 | 19.70 |

## A  SPECIALIZED TASK EVALUATION

We extended MoNE to two specialized tasks: Math and GSM8K. We conducted experiments on both base models and instruct models. To calibrate the instruct models, we adopted the first 100 samples from the training dataset of each task. The results are reported in Tables 4–8.

Our experiments on specialized tasks reveal two important findings regarding the effectiveness of different calibration strategies. After pruning with pretraining data (Zyda2), larger base models preserve significantly more accuracy, which is consistent with our observations from general tasks in Section 5.2. However, pretraining data cannot accurately capture the distribution of specialized tasks, leading to accuracy drops of up to 49% for the smallest model, OLMoE. This demonstrates that while model scale provides some resilience to pruning, the mismatch between pretraining data and

Table 7: Base and Instruct Qwen2-57B-A14B models on Math and GSM8K.

| Model | GSM8K | Math |
|---|---|---|
| *No calibration* | | |
| Qwen2-57B-A14B | 79.68 | 40.50 |
| Qwen2-57B-A14B-Instruct | 69.90 | 31.30 |
| *Calibration: Zyda2* | | |
| Qwen2-57B-A14B | 74.07 | 31.28 |
| Qwen2-57B-A14B-Instruct | 64.90 | 22.84 |
| *Calibration: Task-specific* | | |
| Qwen2-57B-A14B-Instruct (Math) | 58.45 | 29.56 |
| Qwen2-57B-A14B-Instruct (GSM8K) | 52.92 | 29.62 |

Table 8: Base and Instruct Qwen3-30B-A3B models on Math and GSM8K.

| Model | GSM8K | Math |
|---|---|---|
| *No calibration* | | |
| Qwen3-30B-A3B | 85.37 | 50.48 |
| Qwen3-30B-A3B-Instruct | 90.90 | 46.94 |
| *Calibration: Zyda2* | | |
| Qwen3-30B-A3B | 71.72 | 9.44 |
| Qwen3-30B-A3B-Instruct | 78.01 | 9.88 |
| *Calibration: Task-specific* | | |
| Qwen3-30B-A3B-Instruct (Math) | 89.76 | 48.52 |
| Qwen3-30B-A3B-Instruct (GSM8K) | 90.30 | 51.70 |

task-specific distributions poses substantial challenges for maintaining performance on specialized tasks.

In contrast, MoNE can effectively preserve performance by pruning instruct models with training data from the specialized tasks themselves. For example, OLMoE-Instruct incurs only minimal accuracy drops of at most 1% for both tasks when calibrated with task-specific data. This result is not surprising, as state-of-the-art models also rely on specialized fine-tuning approaches such as supervised fine-tuning and reinforcement learning with domain-specific data to enhance their capability in these tasks (Muennighoff et al., 2025; Liu et al., 2025; Qwen, 2024; Yang et al., 2025). The effectiveness of task-specific calibration suggests that the distribution alignment between calibration data and target tasks plays a critical role in determining post-pruning performance.

In summary, we acknowledge that calibration via pretraining data is insufficient to preserve model capability for specialized tasks, and task-specific calibration can effectively mitigate this issue. A promising future direction for model compression is to develop methods that bridge the gap between these two calibration approaches, potentially through hybrid calibration strategies or adaptive data selection techniques that better capture task-specific distributions while maintaining broad coverage.

## B  PRUNING METRIC ABLATION STUDY

We conducted an extended ablation study on the redundancy score to evaluate the sensitivity of MoNE to different scoring formulations. Following the same evaluation configuration as in Section 5.2, we compared MoNE against five variants: normalized scores, log-sum aggregation, and weighted sum with varying emphasis on output variance (25%, 50%, and 75%). The results are reported in Table 9.

The results indicate that the pruning score exhibits minimal scale sensitivity across different formulations. In particular, the log-sum aggregation shows nearly identical results to MoNE, which is expected since $\log(\theta^{var}) + \log(\theta^{freq}) = \log(\theta^{var} \times \theta^{freq})$ does not affect the partial ordering

Table 9: Ablation study on redundancy scoring methods across different models. All values represent average performance scores.

| Model | MoNE | Normalized | Log-sum | Weighted 25% | Weighted 50% | Weighted 75% |
|---|---|---|---|---|---|---|
| OLMoE | 61.04 | 61.39 | 61.23 | 57.07 | 57.07 | 57.07 |
| Moonlight | 69.10 | 68.58 | 69.05 | 68.95 | 69.03 | 69.07 |
| DeepSeek-V2-Lite | 66.12 | 65.70 | 66.02 | 65.63 | 65.66 | 66.07 |
| Qwen2-57B-A14B | 71.75 | 71.73 | 71.75 | 72.55 | 72.57 | 72.53 |
| Qwen3-30B-A3B | 72.69 | 73.62 | 73.66 | 71.96 | 72.29 | 72.48 |

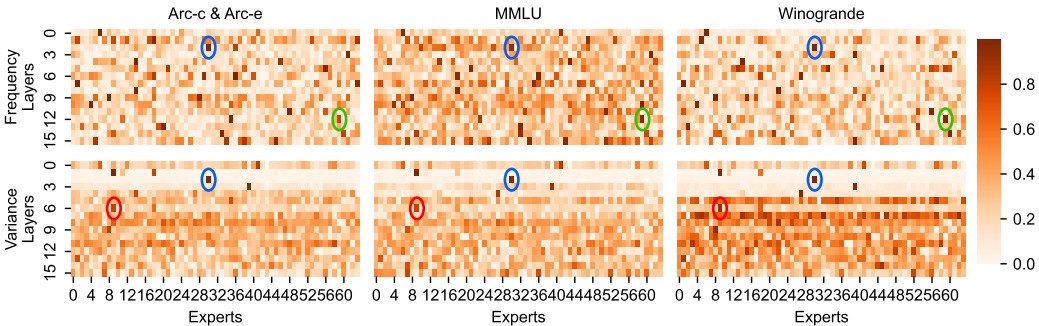

Figure 5: Layer-wise normalized expert access frequency and output variance of OLMoE for Arc-C & Arc-E, MMLU and Winogrande.

during expert redundancy ranking. This mathematical equivalence ensures that both formulations produce the same pruning decisions despite their different representations.

We did not include learnable weights in this ablation study, as training five models with learnable parameters would be prohibitively expensive and time-consuming. However, the weighted sum results reveal model-dependent behavior, where only Qwen2-57B-A14B achieves minor improvements of approximately 0.8% with weighted scoring. In contrast, smaller models like OLMoE incur severe performance drops of around 4% under the weighted sum formulation. This suggests that while larger models may benefit from adjusting the relative importance of variance and frequency components, smaller models are more sensitive to such modifications and perform better with the balanced geometric mean approach used in MoNE.

## C    MORE DETAILED RESULTS

This section presents the experiment results on Zyda2 dataset with 100, 500 and 1000 calibration samples in Table 10, Table 11 and Table 12. Table 13, Table 14 and Table 15 presents the experiment results on C4 dataset with 100, 500 and 1000 calibration samples. The observations are similar to those in Section 5.2 and Section 5.3. Nevertheless, all methods incurs more aggressive performance drop at higher (50%) pruning ratio. Such phenomenon is inevitable for all model compression approaches and inspires the development of MoNE to push the performance limits of structured pruning.

## D    REDUNDANT EXPERT VISUALIZATION

This section complements the visualization of redundant experts for OLMoE, Moonlight and Deepseek-V2-Lite across the nine downstream tasks. The results are depicted in Figure 5 - Figure 12. As mentioned in Figure 1, for each figure, expert in blue circles has both high frequency and variance. Expert in red circles only has high variance. Expert in green circles only has high frequency. For each model across the nine downstream tasks, we can always identify the same important experts, validating the effectiveness of the redundancy metric, i.e., the expert access frequency and output variance.

Table 10: Zero shot performance with 100 calibration samples from Zyda2 dataset. Best results are in **bold**, and the second best are underlined.

(a) OLMoE

| Pruning ratio | Model/Method | Arc-c | Arc-e | BoolQ | COPA | MMLU | OBQA | PIQA | RTE | Winogrande | Average |
|---|---|---|---|---|---|---|---|---|---|---|---|
| 0% | OLMoE | 49.23 | 76.89 | 70.09 | 85.00 | 53.54 | 44.40 | 79.76 | 71.84 | 68.90 | 66.63 |
| 50% | Angular | 27.22 | 37.50 | 53.91 | 62.00 | 23.96 | 26.60 | 58.27 | 52.35 | 51.85 | 43.74 |
| | FLAP | **29.18** | **54.92** | 62.17 | 68.00 | **30.51** | **29.60** | **67.57** | 55.23 | 56.27 | 50.39 |
| | MC-SMoE | 24.49 | 31.44 | 59.33 | 67.00 | 23.01 | 26.00 | 53.92 | 53.43 | 53.12 | 43.53 |
| | RS | 21.50 | 28.62 | 39.45 | 61.00 | 23.27 | 26.00 | 52.34 | 51.99 | 51.85 | 39.56 |
| | MoNE (Ours) | 28.16 | 40.24 | **63.12** | **78.00** | 25.21 | **32.40** | 63.33 | **60.65** | **63.54** | **50.52** |

(b) Moonlight

| Pruning ratio | Model/Method | Arc-c | Arc-e | BoolQ | COPA | MMLU | OBQA | PIQA | RTE | Winogrande | Average |
|---|---|---|---|---|---|---|---|---|---|---|---|
| 0% | Moonlight | 58.28 | 82.49 | 80.40 | 92.00 | 67.30 | 45.60 | 81.12 | 65.70 | 71.11 | 71.56 |
| 50% | Angular | 27.90 | 28.54 | 48.01 | 49.00 | 25.67 | 28.80 | 52.56 | 51.99 | 47.75 | 40.02 |
| | FLAP | 33.87 | 61.36 | 63.30 | 75.00 | **36.80** | 36.00 | 69.37 | 57.04 | 62.12 | 54.98 |
| | MC-SMoE | 29.52 | 47.94 | 59.54 | 79.00 | 23.94 | 31.40 | 67.30 | 57.04 | 60.46 | 50.68 |
| | RS | 37.80 | 58.42 | 70.86 | 89.00 | 23.27 | 38.00 | 78.18 | 57.76 | 70.80 | 58.23 |
| | MoNE (Ours) | **43.09** | **70.03** | **76.12** | 90.00 | 23.57 | **40.80** | **78.78** | 58.84 | 70.17 | **61.27** |

(c) Deepseek-V2-Lite

| Pruning ratio | Model/Method | Arc-c | Arc-e | BoolQ | COPA | MMLU | OBQA | PIQA | RTE | Winogrande | Average |
|---|---|---|---|---|---|---|---|---|---|---|---|
| 0% | Deepseek-V2-Lite | 48.72 | 76.18 | 79.88 | 88.00 | 54.96 | 43.60 | 80.25 | 61.37 | 71.51 | 67.16 |
| 50% | Angular | 24.06 | 32.79 | 40.40 | 61.00 | 23.22 | 26.80 | 56.42 | 57.76 | 49.09 | 41.28 |
| | FLAP | 35.24 | 60.31 | 69.66 | 79.00 | 36.13 | 35.20 | 74.76 | 56.68 | 64.09 | 56.79 |
| | MC-SMoE | 24.57 | 35.82 | 56.36 | 67.00 | 27.40 | 26.00 | 53.91 | 61.37 | 53.91 | 45.23 |
| | RS | 36.01 | 57.45 | 57.98 | **89.00** | 24.91 | **40.80** | 78.02 | 54.15 | 62.75 | 55.67 |
| | MoNE (Ours) | **37.20** | **67.17** | **73.39** | 84.00 | **42.30** | 36.80 | 75.30 | 59.57 | 67.88 | **60.40** |

(d) Qwen2-57B-A14B

| Pruning ratio | Model/Method | Arc-c | Arc-e | BoolQ | COPA | MMLU | OBQA | PIQA | RTE | Winogrande | Avg. |
|---|---|---|---|---|---|---|---|---|---|---|---|
| 0% | Qwen2-57B-A14B | 49.66 | 69.44 | 86.45 | 93.00 | 74.06 | 44.20 | 81.23 | 74.73 | 74.27 | 71.89 |
| 50% | Angular | 25.17 | 31.19 | 45.08 | 54.00 | 23.74 | 29.80 | 54.62 | 53.07 | 50.28 | 40.77 |
| | FLAP | 36.35 | 61.95 | 73.43 | 80.00 | 46.50 | 38.20 | 74.59 | 68.95 | 68.59 | 60.95 |
| | MC-SMoE | 41.64 | 64.73 | 79.91 | **87.00** | 59.65 | 38.80 | 77.53 | **71.84** | 68.27 | 65.49 |
| | RS | 20.90 | 29.38 | 50.83 | 57.00 | 23.39 | 25.40 | 52.88 | 51.26 | 49.25 | 40.03 |
| | MoNE (Ours) | **45.43** | **65.22** | **80.99** | 85.00 | **62.00** | **42.00** | **78.41** | 70.90 | **69.51** | **66.61** |

(e) Qwen3-30B-A3B

| Pruning ratio | Model/Method | Arc-c | Arc-e | BoolQ | COPA | MMLU | OBQA | PIQA | RTE | Winogrande | Avg. |
|---|---|---|---|---|---|---|---|---|---|---|---|
| 0% | Qwen3-30B-A3B | 55.89 | 79.42 | 88.69 | 84.00 | 77.82 | 44.80 | 80.30 | 82.31 | 70.64 | 73.76 |
| 50% | Angular | 26.37 | 39.48 | 50.86 | 59.00 | 24.65 | 28.80 | 60.17 | 55.23 | 48.46 | 43.67 |
| | FLAP | 36.52 | 61.99 | 70.80 | 78.00 | 48.92 | 34.00 | 70.57 | 74.37 | 64.96 | 60.01 |
| | MC-SMoE | 39.76 | 63.09 | 85.84 | 83.00 | 56.30 | 39.40 | 74.54 | **78.34** | 69.14 | 65.49 |
| | RS | **48.04** | **73.06** | **86.61** | 83.00 | **61.37** | **42.40** | **79.43** | 70.40 | 68.27 | **68.06** |
| | MoNE (Ours) | 44.62 | 67.05 | 85.41 | **85.00** | 54.58 | 41.60 | 79.00 | 71.84 | 67.80 | 66.32 |

Table 11: Zero shot performance with 500 calibration samples from Zyda2 dataset. Best results are in **bold**, and the second best are underlined.

(a) OLMoE

| Pruning ratio | Model/Method | Arc-c | Arc-e | BoolQ | COPA | MMLU | OBQA | PIQA | RTE | Winogrande | Average |
|---|---|---|---|---|---|---|---|---|---|---|---|
| 0% | OLMoE | 49.23 | 76.89 | 70.09 | 85.00 | 53.54 | 44.40 | 79.76 | 71.84 | 68.90 | 66.63 |
| 25% | Angular | 32.76 | 61.91 | 61.71 | 74.00 | 23.13 | 37.60 | 71.65 | 53.07 | 55.09 | 52.33 |
| | FLAP | 37.03 | 63.43 | 64.28 | 81.00 | 41.12 | 38.80 | 72.63 | 54.51 | 63.54 | 57.37 |
| | MC-SMoE | 33.36 | 54.46 | 70.03 | 81.00 | 37.05 | 33.80 | 68.17 | 64.62 | 65.19 | 56.41 |
| | RS | 23.81 | 40.91 | 57.92 | 69.00 | 27.79 | 30.20 | 63.71 | 50.18 | 57.38 | 46.77 |
| | MoNE (Ours) | **41.04** | **65.66** | **70.24** | **87.00** | 41.21 | **40.00** | **76.61** | 64.98 | **66.61** | **61.48** |
| 50% | Angular | 27.22 | 37.50 | 53.91 | 62.00 | 23.96 | 26.60 | 58.27 | 52.35 | 51.85 | 43.74 |
| | FLAP | **30.38** | **52.99** | 62.17 | 70.00 | **30.91** | **33.20** | **66.65** | **59.21** | 57.06 | **51.40** |
| | MC-SMoE | 25.43 | 32.28 | 54.80 | 66.00 | 22.95 | 25.40 | 55.22 | 54.51 | 54.14 | 43.41 |
| | RS | 25.34 | 28.24 | 42.48 | 56.00 | 23.07 | 26.40 | 52.34 | 52.35 | 51.78 | 39.78 |
| | MoNE (Ours) | 26.28 | 36.03 | **65.17** | **75.00** | 26.02 | 29.00 | 61.64 | 57.40 | **62.67** | 48.80 |

(b) Moonlight

| Pruning ratio | Model/Method | Arc-c | Arc-e | BoolQ | COPA | MMLU | OBQA | PIQA | RTE | Winogrande | Average |
|---|---|---|---|---|---|---|---|---|---|---|---|
| 0% | Moonlight | 58.28 | 82.49 | 80.40 | 92.00 | 67.30 | 45.60 | 81.12 | 65.70 | 71.11 | 71.56 |
| 25% | Angular | 39.76 | 52.69 | 38.90 | 79.00 | 42.57 | 32.20 | 68.50 | 61.01 | 62.04 | 52.96 |
| | FLAP | 48.55 | 76.05 | 77.49 | 89.00 | **55.12** | 42.40 | 76.66 | **65.34** | 68.59 | 66.58 |
| | MC-SMoE | 37.46 | 63.47 | 76.82 | 81.00 | 48.27 | 35.00 | 71.11 | 58.84 | 70.17 | 60.24 |
| | RS | **55.63** | **79.46** | 78.93 | **91.00** | 46.60 | **45.80** | **80.90** | 59.93 | **72.14** | 67.82 |
| | MoNE (Ours) | 55.03 | 78.96 | **79.36** | 90.00 | 54.39 | 45.40 | 80.69 | 58.48 | 71.74 | **68.23** |
| 50% | Angular | 27.90 | 28.54 | 48.01 | 49.00 | 25.67 | 28.80 | 52.56 | 51.99 | 47.75 | 40.02 |
| | FLAP | 34.98 | 61.32 | 65.14 | 73.00 | **37.93** | 35.60 | 69.53 | 56.68 | 62.35 | 55.17 |
| | MC-SMoE | 22.87 | 29.34 | 58.93 | 72.00 | 23.91 | 26.00 | 55.06 | 54.51 | 52.72 | 43.93 |
| | RS | 37.71 | 59.97 | 71.65 | **89.00** | 25.48 | **38.00** | 76.22 | 57.04 | 68.82 | 58.21 |
| | MoNE (Ours) | **38.23** | **64.48** | **75.90** | 87.00 | 23.89 | 39.20 | **77.42** | 58.48 | 70.56 | **59.46** |

(c) Deepseek-V2-Lite

| Pruning ratio | Model/Method | Arc-c | Arc-e | BoolQ | COPA | MMLU | OBQA | PIQA | RTE | Winogrande | Average |
|---|---|---|---|---|---|---|---|---|---|---|---|
| 0% | Deepseek-V2-Lite | 48.72 | 76.18 | 79.88 | 88.00 | 54.96 | 43.60 | 80.25 | 61.37 | 71.51 | 67.16 |
| 25% | Angular | 32.00 | 53.28 | 64.92 | 75.00 | 26.95 | 34.00 | 71.33 | 58.84 | 61.01 | 53.04 |
| | FLAP | 43.86 | 72.18 | 75.93 | 85.00 | 47.22 | 41.80 | 78.45 | **62.09** | 68.27 | 63.87 |
| | MC-SMoE | 33.53 | 52.95 | 73.67 | 81.00 | 41.68 | 32.20 | 66.70 | 52.35 | 70.17 | 56.03 |
| | RS | **48.98** | **73.23** | 71.77 | **89.00** | 52.68 | **44.60** | 79.33 | 61.01 | 70.32 | **65.66** |
| | MoNE (Ours) | 44.62 | 73.11 | **78.01** | **90.00** | 48.29 | 41.80 | **79.43** | 59.21 | **71.35** | 65.09 |
| 50% | Angular | 24.06 | 32.79 | 40.40 | 61.00 | 23.22 | 26.80 | 56.42 | 57.76 | 49.09 | 41.28 |
| | FLAP | 33.11 | 60.73 | 67.77 | 78.00 | 31.87 | 36.80 | 72.91 | 55.23 | 63.93 | 55.59 |
| | MC-SMoE | 25.34 | 32.07 | 47.98 | 59.00 | 26.28 | 25.60 | 56.20 | 54.51 | 53.75 | 42.30 |
| | RS | **38.14** | 62.42 | 53.03 | **86.00** | 38.98 | **38.80** | 74.37 | 48.01 | 64.64 | 56.04 |
| | MoNE (Ours) | 36.69 | **66.04** | **73.36** | **86.00** | **41.07** | 35.60 | **75.73** | 58.12 | 69.69 | **60.26** |

Table 12: Zero shot performance with 1000 calibration samples from Zyda2 dataset. Best results are in **bold**, and the second best are underlined.

(a) OLMoE

| Pruning ratio | Model/Method | Arc-c | Arc-e | BoolQ | COPA | MMLU | OBQA | PIQA | RTE | Winogrande | Average |
|---|---|---|---|---|---|---|---|---|---|---|---|
| 0% | OLMoE | 49.23 | 76.89 | 70.09 | 85.00 | 53.54 | 44.40 | 79.76 | 71.84 | 68.90 | 66.63 |
| 25% | Angular | 32.76 | 61.91 | 61.71 | 74.00 | 23.13 | 37.60 | 71.65 | 53.07 | 55.09 | 52.33 |
| | FLAP | 38.91 | **66.20** | 64.65 | 79.00 | 40.05 | 37.60 | 74.65 | 62.82 | 63.77 | 58.63 |
| | MC-SMoE | 38.31 | 61.66 | 61.87 | 73.00 | 33.85 | 33.20 | 66.38 | 57.04 | 65.43 | 54.53 |
| | RS | 26.62 | 43.35 | 59.76 | 70.00 | 27.50 | 30.00 | 65.02 | 49.46 | 56.67 | 47.60 |
| | MoNE (Ours) | **42.32** | 64.52 | **66.45** | **88.00** | **41.55** | **40.80** | **78.02** | **64.26** | **67.01** | **61.44** |
| 50% | Angular | 27.22 | 37.50 | 53.91 | 62.00 | 23.96 | 26.60 | 58.27 | 52.35 | 51.85 | 43.74 |
| | FLAP | **30.46** | **54.88** | 62.32 | 70.00 | **29.63** | **32.00** | **67.95** | **57.04** | 57.54 | **51.31** |
| | MC-SMoE | 25.43 | 35.31 | 55.11 | 67.00 | 22.92 | 25.40 | 54.52 | 51.99 | 51.14 | 43.20 |
| | RS | 24.23 | 29.50 | 41.44 | 58.00 | 23.45 | 24.20 | 51.25 | 50.54 | 51.07 | 39.30 |
| | MoNE (Ours) | 26.71 | 37.21 | **65.44** | **75.00** | 26.50 | 31.80 | 63.22 | 55.96 | **63.46** | 49.48 |

(b) Moonlight

| Pruning ratio | Model/Method | Arc-c | Arc-e | BoolQ | COPA | MMLU | OBQA | PIQA | RTE | Winogrande | Average |
|---|---|---|---|---|---|---|---|---|---|---|---|
| 0% | Moonlight | 58.28 | 82.49 | 80.40 | 92.00 | 67.30 | 45.60 | 81.12 | 65.70 | 71.11 | 71.56 |
| 25% | Angular | 39.76 | 52.69 | 38.90 | 79.00 | 42.57 | 32.20 | 68.50 | 61.01 | 62.04 | 52.96 |
| | FLAP | 49.49 | 76.52 | 76.91 | **90.00** | 54.47 | 42.00 | 77.26 | **65.70** | 69.06 | 66.82 |
| | MC-SMoE | 34.81 | 57.74 | 77.22 | 85.00 | 36.13 | 34.60 | 71.49 | 58.12 | 71.35 | 58.50 |
| | RS | **56.23** | 79.25 | 79.02 | **91.00** | 46.76 | 45.20 | 80.63 | 59.57 | **72.14** | 67.76 |
| | MoNE (Ours) | 55.55 | **79.29** | **79.66** | **90.00** | **54.79** | **45.60** | **80.69** | 59.21 | **72.14** | **68.55** |
| 50% | Angular | 27.90 | 28.54 | 48.01 | 49.00 | 25.67 | 28.80 | 52.56 | 51.99 | 47.75 | 40.02 |
| | FLAP | 36.95 | 63.09 | 64.56 | 71.00 | **37.14** | 34.80 | 69.86 | 57.40 | 63.22 | 55.33 |
| | MC-SMoE | 23.12 | 41.04 | 61.83 | 68.00 | 25.44 | 26.80 | 58.43 | 53.07 | 53.75 | 45.72 |
| | RS | **39.42** | 64.56 | 70.98 | **90.00** | 24.76 | **40.60** | 77.58 | **58.12** | 68.98 | 59.45 |
| | MoNE (Ours) | **39.42** | **65.82** | **76.15** | 88.00 | 23.81 | 40.40 | **78.40** | **58.12** | **70.88** | **60.11** |

(c) Deepseek-V2-Lite

| Pruning ratio | Model/Method | Arc-c | Arc-e | BoolQ | COPA | MMLU | OBQA | PIQA | RTE | Winogrande | Average |
|---|---|---|---|---|---|---|---|---|---|---|---|
| 0% | Deepseek-V2-Lite | 48.72 | 76.18 | 79.88 | 88.00 | 54.96 | 43.60 | 80.25 | 61.37 | 71.51 | 67.16 |
| 25% | Angular | 32.00 | 53.28 | 64.92 | 75.00 | 26.95 | 34.00 | 71.33 | 58.84 | 61.01 | 53.04 |
| | FLAP | 44.28 | 72.39 | 76.12 | 85.00 | 47.16 | 41.20 | 78.29 | **62.82** | 67.40 | 63.85 |
| | MC-SMoE | 39.33 | 64.31 | 71.53 | 85.00 | 41.84 | 40.00 | 76.12 | 58.48 | 69.53 | 60.68 |
| | RS | **49.23** | 73.32 | 71.38 | **89.00** | **52.44** | **45.00** | **79.71** | 60.29 | **71.19** | **65.73** |
| | MoNE (Ours) | 44.71 | **73.99** | **78.35** | 88.00 | 49.19 | 42.20 | 79.27 | 59.93 | 70.88 | 65.17 |
| 50% | Angular | 24.06 | 32.79 | 40.40 | 61.00 | 23.22 | 26.80 | 56.42 | **57.76** | 49.09 | 41.28 |
| | FLAP | 32.59 | 60.77 | 68.13 | 77.00 | 31.43 | 36.80 | 73.94 | 54.87 | 61.64 | 55.24 |
| | MC-SMoE | 33.19 | 47.81 | 55.35 | 81.00 | 24.91 | 33.80 | 67.52 | 51.99 | 64.33 | 51.10 |
| | RS | 37.12 | 61.95 | 54.53 | **88.00** | 39.15 | **38.40** | **75.46** | 51.62 | 62.98 | 56.58 |
| | MoNE (Ours) | **37.71** | **65.36** | **73.49** | 84.00 | **41.22** | 36.60 | 75.30 | 57.40 | **69.53** | **60.07** |

Table 13: Zero shot performance with 100 calibration samples from C4 dataset. Best results are in **bold**, and the second best are underlined.

(a) OLMoE

| Pruning ratio | Model/Method | Arc-c | Arc-e | BoolQ | COPA | MMLU | OBQA | PIQA | RTE | Winogrande | Average |
|---|---|---|---|---|---|---|---|---|---|---|---|
| 0% | OLMoE | 49.23 | 76.89 | 70.09 | 85.00 | 53.54 | 44.40 | 79.76 | 71.84 | 68.90 | 66.63 |
| 25% | Angular | 32.76 | 61.91 | 61.71 | 74.00 | 23.13 | 37.60 | 71.65 | 53.07 | 55.09 | 52.33 |
| | FLAP | 36.01 | **62.67** | 64.83 | 75.00 | **36.31** | 37.00 | 75.73 | **58.84** | 62.75 | 56.57 |
| | MC-SMoE | 32.76 | 51.05 | 54.71 | 80.00 | 23.05 | 37.80 | 70.89 | 53.43 | 66.61 | 52.26 |
| | RS | 34.56 | 50.38 | 63.64 | 85.00 | 26.51 | 39.60 | 76.01 | 55.96 | 64.01 | 55.07 |
| | MoNE (Ours) | **36.69** | 55.18 | **67.03** | **86.00** | 25.53 | **40.20** | **77.80** | 56.68 | **67.64** | **56.97** |
| 50% | Angular | **27.22** | 37.50 | 53.91 | 62.00 | 23.96 | 26.60 | 58.27 | 52.35 | 51.85 | 43.74 |
| | FLAP | 26.62 | **50.51** | 62.14 | 62.00 | **26.32** | 28.80 | **69.86** | 52.71 | 57.06 | 48.45 |
| | MC-SMoE | 25.68 | 32.24 | 57.65 | 60.00 | 22.83 | 27.40 | 58.60 | 52.35 | 51.38 | 43.12 |
| | RS | 23.12 | 31.99 | 45.02 | 59.00 | 23.80 | 24.80 | 53.59 | 51.62 | 49.57 | 40.28 |
| | MoNE (Ours) | 26.37 | 39.65 | **62.29** | 75.00 | 22.93 | **32.00** | 66.10 | **52.71** | **61.01** | **48.67** |

(b) Moonlight

| Pruning ratio | Model/Method | Arc-c | Arc-e | BoolQ | COPA | MMLU | OBQA | PIQA | RTE | Winogrande | Average |
|---|---|---|---|---|---|---|---|---|---|---|---|
| 0% | Moonlight | 58.28 | 82.49 | 80.40 | 92.00 | 67.30 | 45.60 | 81.12 | 65.70 | 71.11 | 71.56 |
| 25% | Angular | 39.76 | 52.69 | 38.90 | 79.00 | 42.57 | 32.20 | 68.50 | 61.01 | 62.04 | 52.96 |
| | FLAP | 46.76 | 75.04 | 75.90 | 85.00 | 52.98 | 41.00 | 78.07 | **65.70** | 67.40 | 65.32 |
| | MC-SMoE | 48.38 | 74.20 | 78.69 | **92.00** | 50.41 | 44.20 | 81.18 | 54.87 | 71.19 | 66.12 |
| | RS | **55.97** | **79.76** | **78.93** | 91.00 | 52.74 | **46.20** | **81.39** | 59.21 | **72.38** | **68.62** |
| | MoNE (Ours) | 54.27 | 79.25 | 78.07 | 90.00 | **53.78** | 45.60 | 81.34 | 57.76 | 72.14 | 68.02 |
| 50% | Angular | 27.90 | 28.54 | 48.01 | 49.00 | 25.67 | 28.80 | 52.56 | 51.99 | 47.75 | 40.02 |
| | FLAP | 31.14 | 54.63 | 62.69 | 73.00 | **32.26** | 30.40 | 70.40 | 58.12 | 60.93 | 52.62 |
| | MC-SMoE | 29.10 | 52.23 | 57.22 | 85.00 | 22.92 | 36.40 | 71.71 | 52.71 | 63.54 | 52.31 |
| | RS | **37.80** | **58.42** | 70.86 | 89.00 | 23.27 | **38.00** | 78.18 | 57.76 | **70.80** | 58.23 |
| | MoNE (Ours) | 33.87 | 57.07 | **72.75** | 90.00 | 22.97 | 38.80 | **78.62** | **61.01** | 70.17 | **58.36** |

(c) Deepseek-V2-Lite

| Pruning ratio | Model/Method | Arc-c | Arc-e | BoolQ | COPA | MMLU | OBQA | PIQA | RTE | Winogrande | Average |
|---|---|---|---|---|---|---|---|---|---|---|---|
| 0% | Deepseek-V2-Lite | 48.72 | 76.18 | 79.88 | 88.00 | 54.96 | 43.60 | 80.25 | 61.37 | 71.51 | 67.16 |
| 25% | Angular | 32.00 | 53.28 | 64.92 | 75.00 | 26.95 | 34.00 | 71.33 | 58.84 | 61.01 | 53.04 |
| | FLAP | 39.68 | 66.58 | 76.45 | 84.00 | 36.14 | 39.80 | 78.67 | 57.04 | 67.48 | 60.65 |
| | MC-SMoE | 36.52 | 59.30 | 59.76 | 81.00 | 36.80 | 37.20 | 75.30 | 54.87 | 69.22 | 56.66 |
| | RS | **49.32** | 74.41 | 69.39 | 90.00 | **50.35** | 43.80 | 80.14 | 62.09 | 70.24 | 65.53 |
| | MoNE (Ours) | 47.44 | 74.03 | **77.71** | 90.00 | 49.10 | 42.60 | 80.25 | 63.18 | 70.24 | **66.06** |
| 50% | Angular | 24.06 | 32.79 | 40.40 | 61.00 | 23.22 | 26.80 | 56.42 | 57.76 | 49.09 | 41.28 |
| | FLAP | 29.95 | 52.36 | **68.72** | 75.00 | 23.45 | 34.20 | 75.46 | 52.71 | 61.09 | 52.55 |
| | MC-SMoE | 28.75 | 36.57 | 59.45 | 82.00 | 23.65 | 30.60 | 64.96 | 51.62 | 59.75 | 48.60 |
| | RS | **36.01** | **57.45** | 57.98 | 89.00 | 24.91 | 40.80 | 78.02 | 54.15 | 62.75 | 55.67 |
| | MoNE (Ours) | 35.24 | 55.47 | 68.20 | 87.00 | 22.94 | 35.40 | 76.50 | 53.43 | 68.35 | 55.84 |

Table 14: Zero shot performance with 500 calibration samples from C4 dataset. Best results are in **bold**, and the second best are underlined.

(a) OLMoE

| Pruning ratio | Model/Method | Arc-c | Arc-e | BoolQ | COPA | MMLU | OBQA | PIQA | RTE | Winogrande | Average |
|---|---|---|---|---|---|---|---|---|---|---|---|
| 0% | OLMoE | 49.23 | 76.89 | 70.09 | 85.00 | 53.54 | 44.40 | 79.76 | 71.84 | 68.90 | 66.63 |
| 25% | Angular | 32.76 | 61.91 | 61.71 | 74.00 | 23.13 | 37.60 | 71.65 | 53.07 | 55.09 | 52.33 |
| | FLAP | 36.26 | **63.80** | 63.49 | 72.00 | 35.09 | 36.00 | 75.35 | 53.07 | 61.96 | 55.23 |
| | MC-SMoE | 26.71 | 48.61 | **65.90** | 69.00 | **35.14** | 31.00 | 61.92 | 54.87 | 61.09 | 50.47 |
| | RS | 34.56 | 49.41 | 63.85 | **86.00** | 27.23 | 39.80 | 75.57 | 56.32 | 64.40 | 55.24 |
| | MoNE (Ours) | **37.88** | 55.89 | 65.23 | **86.00** | 25.03 | **41.20** | 77.42 | 57.40 | 67.80 | **57.10** |
| 50% | Angular | 27.22 | 37.50 | 53.91 | 62.00 | 23.96 | 26.60 | 58.27 | 52.35 | 51.85 | 43.74 |
| | FLAP | 26.71 | **50.42** | 62.20 | 64.00 | **27.52** | 30.20 | 70.29 | 52.35 | 55.09 | 48.75 |
| | MC-SMoE | 23.98 | 32.07 | 61.77 | 63.00 | 23.01 | 25.40 | 54.79 | **57.40** | 52.49 | 43.77 |
| | RS | 22.95 | 31.78 | 49.11 | 59.00 | 23.75 | 23.00 | 53.59 | 53.43 | 50.04 | 40.74 |
| | MoNE (Ours) | **29.01** | 42.51 | **62.29** | 79.00 | 22.98 | **32.00** | 70.84 | 52.71 | **60.06** | **50.16** |

(b) Moonlight

| Pruning ratio | Model/Method | Arc-c | Arc-e | BoolQ | COPA | MMLU | OBQA | PIQA | RTE | Winogrande | Average |
|---|---|---|---|---|---|---|---|---|---|---|---|
| 0% | Moonlight | 58.28 | 82.49 | 80.40 | 92.00 | 67.30 | 45.60 | 81.12 | 65.70 | 71.11 | 71.56 |
| 25% | Angular | 39.76 | 52.69 | 38.90 | 79.00 | 42.57 | 32.20 | 68.50 | 61.01 | 62.04 | 52.96 |
| | FLAP | 45.22 | 73.40 | 72.29 | 84.00 | **53.81** | 40.40 | 78.13 | **64.98** | 67.32 | 64.40 |
| | MC-SMoE | 48.38 | 74.16 | 78.56 | 89.00 | 48.78 | 43.00 | 81.12 | 58.48 | 71.03 | 65.84 |
| | RS | **55.72** | **79.80** | **79.20** | **91.00** | 52.93 | 44.60 | **81.23** | 59.57 | **71.74** | **68.42** |
| | MoNE (Ours) | 54.52 | 78.87 | 78.56 | 90.00 | 53.80 | **45.20** | 81.07 | 57.04 | **71.74** | 67.87 |
| 50% | Angular | 27.90 | 28.54 | 48.01 | 49.00 | 25.67 | 28.80 | 52.56 | 51.99 | 47.75 | 40.02 |
| | FLAP | 31.83 | 54.50 | 63.18 | 74.00 | **29.18** | 31.40 | 71.16 | 55.96 | 61.88 | 52.57 |
| | MC-SMoE | 33.45 | 54.59 | 63.82 | 79.00 | 23.95 | 33.20 | 72.31 | 52.71 | 61.09 | 52.68 |
| | RS | **39.59** | **58.67** | 70.49 | 86.00 | 23.36 | **38.60** | 78.02 | 54.15 | 70.24 | 57.68 |
| | MoNE (Ours) | 33.28 | 56.40 | **71.96** | **91.00** | 22.95 | 37.60 | **78.73** | **58.12** | 70.64 | **57.85** |

(c) Deepseek-V2-Lite

| Pruning ratio | Model/Method | Arc-c | Arc-e | BoolQ | COPA | MMLU | OBQA | PIQA | RTE | Winogrande | Average |
|---|---|---|---|---|---|---|---|---|---|---|---|
| 0% | Deepseek-V2-Lite | 48.72 | 76.18 | 79.88 | 88.00 | 54.96 | 43.60 | 80.25 | 61.37 | 71.51 | 67.16 |
| 25% | Angular | 32.00 | 53.28 | 64.92 | 75.00 | 26.95 | 34.00 | 71.33 | 58.84 | 61.01 | 53.04 |
| | FLAP | 40.53 | 64.81 | **77.43** | 84.00 | 37.70 | 39.60 | 79.43 | 57.04 | 67.32 | 60.87 |
| | MC-SMoE | 39.33 | 61.99 | 60.95 | 84.00 | 41.37 | 38.60 | 77.75 | 58.48 | 62.04 | 58.28 |
| | RS | **49.66** | **74.49** | 65.63 | **91.00** | **50.04** | **43.60** | **79.92** | 61.01 | 69.77 | **65.01** |
| | MoNE (Ours) | 46.59 | 73.19 | 77.37 | 89.00 | 48.80 | 43.40 | 79.87 | **62.09** | 70.80 | 65.68 |
| 50% | Angular | 24.06 | 32.79 | 40.40 | 61.00 | 23.22 | 26.80 | 56.42 | **57.76** | 49.09 | 41.28 |
| | FLAP | 30.97 | 52.27 | **68.47** | 77.00 | 23.16 | 36.40 | 75.08 | 52.71 | 62.75 | 53.20 |
| | MC-SMoE | 23.55 | 35.73 | 57.43 | 71.00 | 23.86 | 32.20 | 61.10 | 51.62 | 51.30 | 45.31 |
| | RS | **36.95** | **58.71** | 51.16 | **87.00** | **24.23** | **40.20** | **77.64** | 54.15 | 61.72 | 54.64 |
| | MoNE (Ours) | 34.39 | 55.72 | 67.68 | 84.00 | 22.90 | 35.60 | 76.28 | 54.51 | **67.72** | **55.42** |

Table 15: Zero shot performance with 1000 calibration samples from C4 dataset. Best results are in **bold**, and the second best are underlined.

(a) OLMoE

| Pruning ratio | Model/Method | Arc-c | Arc-e | BoolQ | COPA | MMLU | OBQA | PIQA | RTE | Winogrande | Average |
|---|---|---|---|---|---|---|---|---|---|---|---|
| 0% | OLMoE | 49.23 | 76.89 | 70.09 | 85.00 | 53.54 | 44.40 | 79.76 | 71.84 | 68.90 | 66.63 |
| 25% | Angular | 32.76 | 61.91 | 61.71 | 74.00 | 23.13 | 37.60 | 71.65 | 53.07 | 55.09 | 52.33 |
| | FLAP | 36.35 | 61.24 | 64.50 | 72.00 | **37.08** | 36.00 | 75.41 | 57.40 | 62.59 | 55.84 |
| | MC-SMoE | 40.10 | **68.22** | 61.65 | 70.00 | 36.91 | 39.60 | 71.65 | **59.57** | 57.93 | 56.18 |
| | RS | 34.39 | 49.41 | 64.89 | 85.00 | 26.64 | 40.60 | 75.95 | 57.04 | 63.46 | 55.26 |
| | MoNE (Ours) | 38.74 | 57.45 | **65.57** | **86.00** | 24.59 | **41.80** | **77.42** | 56.68 | **68.43** | **57.41** |
| 50% | Angular | 27.22 | 37.50 | 53.91 | 62.00 | 23.96 | 26.60 | 58.27 | 52.35 | 51.85 | 43.74 |
| | FLAP | 26.79 | **49.92** | 62.17 | 63.00 | **27.34** | 31.60 | 70.24 | 53.43 | 56.35 | 48.98 |
| | MC-SMoE | 26.96 | 41.58 | 58.56 | 56.00 | 23.00 | 28.60 | 60.23 | 52.35 | 50.83 | 44.23 |
| | RS | 23.81 | 32.28 | 46.09 | 58.00 | 23.76 | 24.00 | 53.59 | 51.99 | 48.86 | 40.26 |
| | MoNE (Ours) | **29.78** | 42.80 | **62.32** | **80.00** | 22.99 | **33.60** | **71.49** | 52.71 | **60.06** | **50.64** |

(b) Moonlight

| Pruning ratio | Model/Method | Arc-c | Arc-e | BoolQ | COPA | MMLU | OBQA | PIQA | RTE | Winogrande | Average |
|---|---|---|---|---|---|---|---|---|---|---|---|
| 0% | Moonlight | 58.28 | 82.49 | 80.40 | 92.00 | 67.30 | 45.60 | 81.12 | 65.70 | 71.11 | 71.56 |
| 25% | Angular | 39.76 | 52.69 | 38.90 | 79.00 | 42.57 | 32.20 | 68.50 | 61.01 | 62.04 | 52.96 |
| | FLAP | 45.22 | 74.20 | 72.78 | 86.00 | 53.40 | 41.20 | 78.35 | **65.34** | 68.90 | 65.04 |
| | MC-SMoE | 51.62 | 76.52 | 77.86 | **92.00** | 43.08 | 43.60 | 80.25 | 55.60 | 71.11 | 65.74 |
| | RS | **55.97** | **79.59** | **78.75** | 90.00 | 53.01 | **45.20** | 81.34 | 59.57 | **71.90** | **68.37** |
| | MoNE (Ours) | 53.75 | 79.38 | 78.53 | 90.00 | **53.75** | **45.80** | **81.39** | 57.76 | 71.59 | 67.99 |
| 50% | Angular | 27.90 | 28.54 | 48.01 | 49.00 | 25.67 | 28.80 | 52.56 | 51.99 | 47.75 | 40.02 |
| | FLAP | 31.31 | 55.26 | 63.73 | 72.00 | **32.98** | 33.00 | 71.16 | 56.32 | 62.75 | 53.17 |
| | MC-SMoE | 28.75 | 49.20 | 58.90 | 75.00 | 23.72 | 33.40 | 66.32 | 53.79 | 59.75 | 49.87 |
| | RS | **38.74** | **58.88** | 69.85 | 88.00 | 23.32 | **38.00** | 77.80 | **61.01** | 70.17 | **58.42** |
| | MoNE (Ours) | 34.81 | 57.58 | **71.96** | **92.00** | 22.95 | 38.60 | **78.18** | 57.04 | 70.72 | 58.20 |

(c) Deepseek-V2-Lite

| Pruning ratio | Model/Method | Arc-c | Arc-e | BoolQ | COPA | MMLU | OBQA | PIQA | RTE | Winogrande | Average |
|---|---|---|---|---|---|---|---|---|---|---|---|
| 0% | Deepseek-V2-Lite | 48.72 | 76.18 | 79.88 | 88.00 | 54.96 | 43.60 | 80.25 | 61.37 | 71.51 | 67.16 |
| 25% | Angular | 32.00 | 53.28 | 64.92 | 75.00 | 26.95 | 34.00 | 71.33 | 58.84 | 61.01 | 53.04 |
| | FLAP | 39.33 | 66.50 | 76.33 | 84.00 | 38.14 | 40.60 | 78.84 | 59.93 | 67.88 | 61.28 |
| | MC-SMoE | 37.29 | 58.42 | 65.72 | 81.00 | 37.91 | 39.00 | 76.12 | 55.60 | 67.80 | 57.65 |
| | RS | **49.23** | **74.33** | 68.44 | 88.00 | **51.59** | **44.40** | **80.25** | 62.45 | 69.69 | **65.38** |
| | MoNE (Ours) | 45.99 | 72.98 | **77.43** | **89.00** | 48.90 | 42.60 | 79.98 | 61.01 | **71.59** | 65.50 |
| 50% | Angular | 24.06 | 32.79 | 40.40 | 61.00 | 23.22 | 26.80 | 56.42 | **57.76** | 49.09 | 41.28 |
| | FLAP | 31.66 | 53.41 | **69.36** | 75.00 | 23.04 | 35.40 | 75.41 | 52.71 | 62.51 | 53.17 |
| | MC-SMoE | 28.16 | 36.49 | 56.94 | 81.00 | **25.76** | 31.80 | 64.69 | 52.71 | 58.09 | 48.40 |
| | RS | **39.08** | **61.32** | 52.54 | 83.00 | 24.58 | 39.40 | **77.86** | 54.51 | 63.46 | **55.08** |
| | MoNE (Ours) | 34.04 | 55.51 | 68.26 | **85.00** | 22.92 | 36.40 | 76.44 | 54.15 | **66.61** | 55.48 |

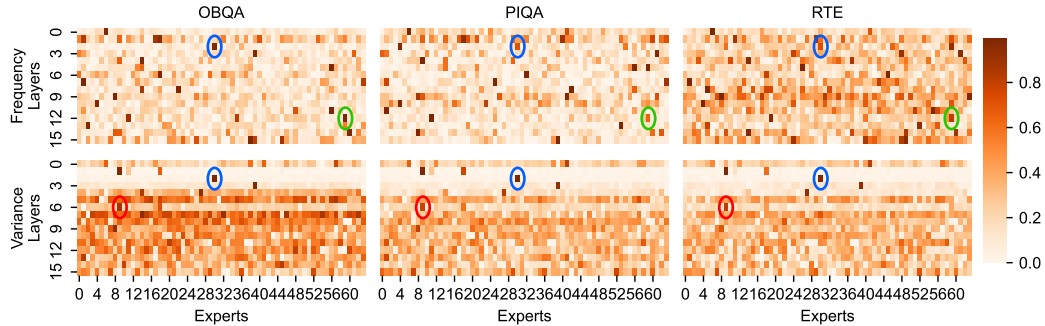

Figure 6: Layer-wise normalized expert access frequency and output variance of OLMoE for OBQA, PIQA and RTE.

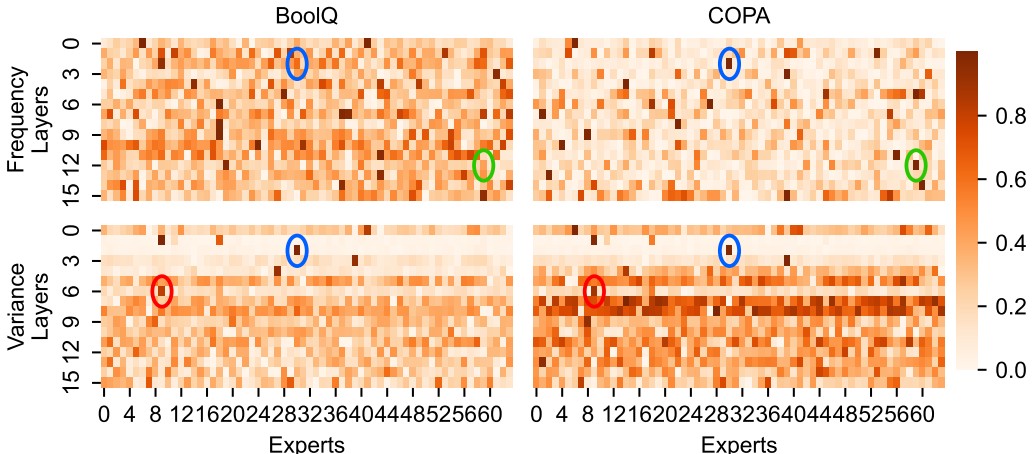

Figure 7: Layer-wise normalized expert access frequency and output variance of OLMoE for BoolQ and COPA.

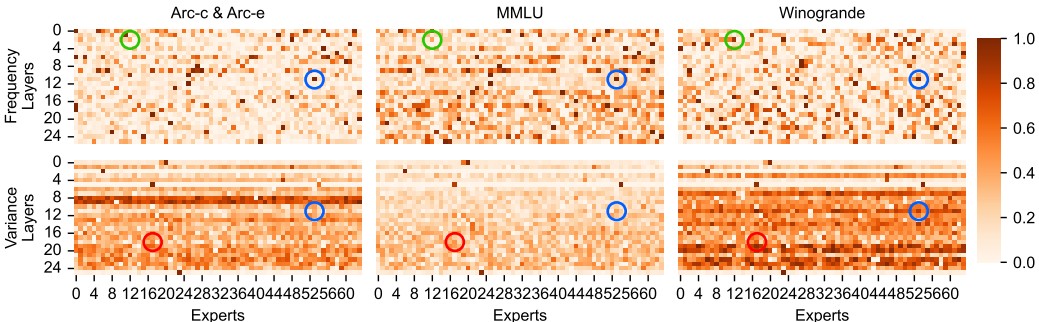

Figure 8: Layer-wise normalized expert access frequency and output variance of Moonlight for Arc-C & Arc-E, MMLU and Winogrande.

# E    COMPREHENSIVE ABLATION STUDY RESULTS

Figure 13 reports the detailed ablation study on the impacts of the three factors: expert access frequency, output variance and novice replacement. The results in this figure validates that the three factors play an important role in maintaining the effectiveness of the pruned models on different model architectures, calibration data sources and calibration sample sizes. Fusing the three factors ensures the robustness of the proposed MoNE.

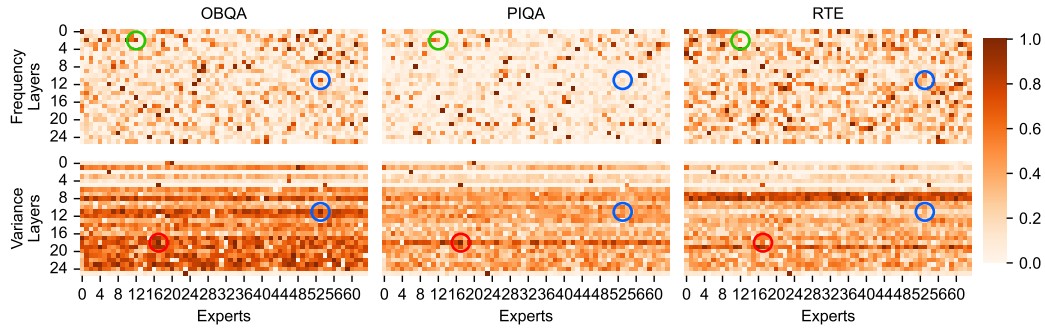

Figure 9: Layer-wise normalized expert access frequency and output variance of Moonlight for OBQA, PIQA and RTE.

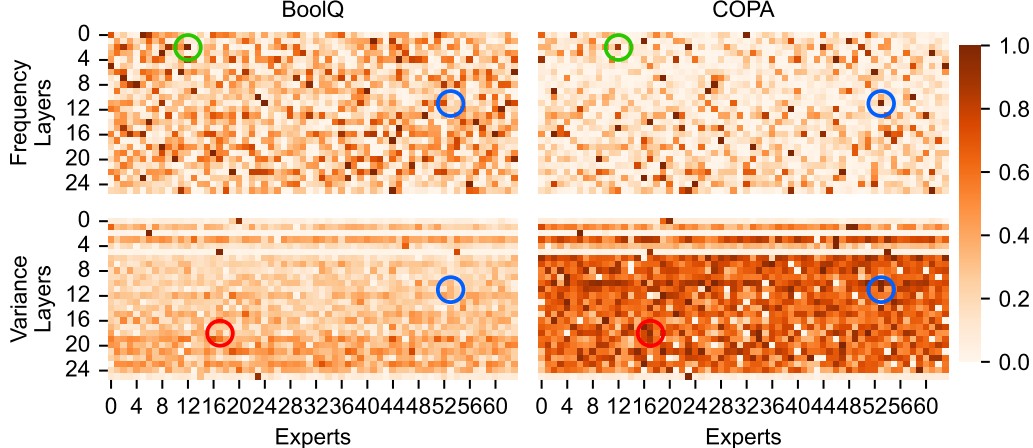

Figure 10: Layer-wise normalized expert access frequency and output variance of Moonlight for BoolQ and COPA.

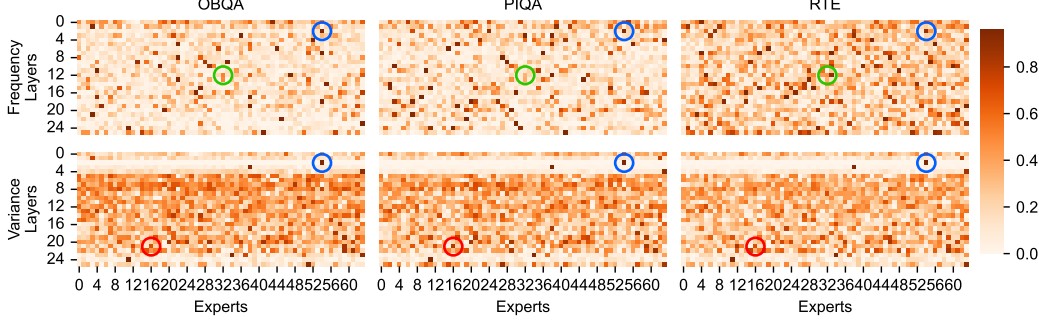

Figure 11: Layer-wise normalized expert access frequency and output variance of Deepseek-V2-Lite for OBQA, PIQA and RTE.

## F  INFERENCE LATENCY AND MEMORY FOOTPRINT

While this work mainly focuses on enhancing performance preserving capability of structured pruning, this section evaluates the inference latency and memory footprint for pruned models. We integrated the pruned Qwen3-30B-A3B with the popular inference framework, SGLang with transformers backend. We fixed the 512 input tokens and 256 output tokens, and profiled the latency and memory footprint with SGLang built-in utilities. We varied the random seed to generate different input sequences. Experiments were conducted on a single H20 GPU. The results are listed in Table 16.

According to Table 16, we would like to clarify three points regarding the latency and memory usage. First, inference latency speedup of MoNE is sensitive to batch size. When batch size is 1, there is minor speedup, as the decoding phase only executes one tokens per step, leading to a memory bound

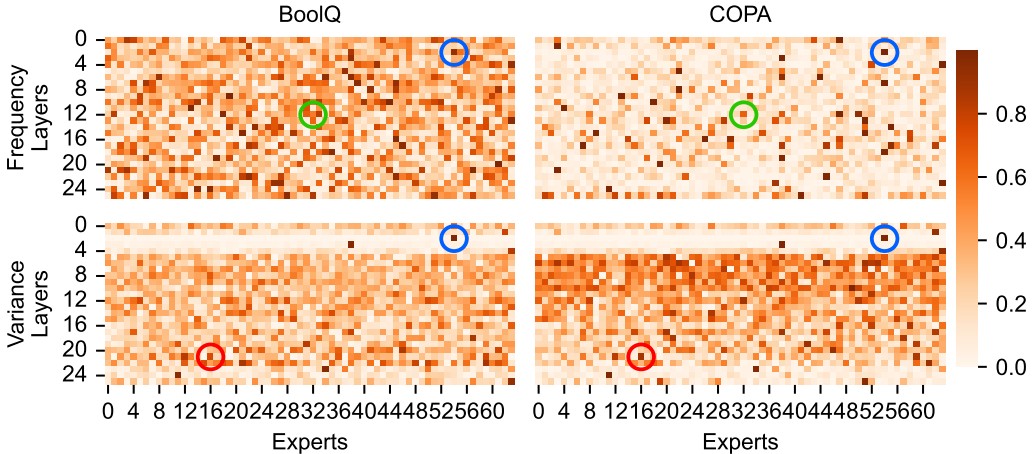

Figure 12: Layer-wise normalized expert access frequency and output variance of Deepseek-V2-Lite for BoolQ and COPA.

Table 16: Inference latency and memory footprint under different pruning ratio, input sequence and batch size.

| Pruning ratio | Seed | Batch size | Novice hit ratios* | Total latency (s) | Speedup | Memory usage (GB) |
|---|---|---|---|---|---|---|
| 0% | 1020 | 1 | 0 | 28.86 | 1 | 62.72 |
| 25% | 1020 | 1 | 0.08 | 27.84 | 1.04 | 47.72 |
| 50% | 1020 | 1 | 0.35 | 26.27 | 1.10 | 32.73 |
| 0% | 1998 | 1 | 0 | 29.01 | 1 | 62.72 |
| 25% | 1998 | 1 | 0.06 | 28.61 | 1.01 | 47.72 |
| 50% | 1998 | 1 | 0.34 | 26.22 | 1.11 | 32.73 |
| 0% | 1020 | 128 | 0 | 183.12 | 1 | 66.00 |
| 25% | 1020 | 128 | 0.08 | 174.60 | 1.05 | 51.00 |
| 50% | 1020 | 128 | 0.34 | 141.07 | 1.30 | 36.01 |
| 0% | 1998 | 128 | 0 | 182.37 | 1 | 66.00 |
| 25% | 1998 | 128 | 0.08 | 169.66 | 1.07 | 51.00 |
| 50% | 1998 | 128 | 0.33 | 142.38 | 1.28 | 36.01 |
| 0% | 1020 | 512 | 0 | 248.02 | 1 | 75.90 |
| 25% | 1020 | 512 | 0.09 | 212.40 | 1.17 | 60.90 |
| 50% | 1020 | 512 | 0.33 | 189.57 | 1.31 | 45.91 |
| 0% | 1998 | 512 | 0 | 245.42 | 1 | 75.90 |
| 25% | 1998 | 512 | 0.08 | 212.88 | 1.15 | 60.90 |
| 50% | 1998 | 512 | 0.34 | 180.92 | 1.36 | 45.91 |

*Novice hit ratios: the portion of tokens routed to novices across the model.

situation where the GPU tensor core is underutilized and the major bottleneck is the GPU memory bandwidth (Hong et al., 2023; Frantar et al., 2025). By increasing the batch size to 512, MoNE can achieve speedup of 36%. Second, inference latency speedup is sensitive to novice hit ratio instead of novice counts (pruning ratio). Qwen3-30B-A3B has 128 experts per layer, and each input token is routed to 8 experts per novices. The actual speedup is decided by the routing results for each input token at runtime. With higher novice hit ratio, MoNE is expected to attain more speedup. While higher pruning ratio may lead to more novices in each layer, it cannot directly translate to speedup. For the most extreme case where there is no token routed to novices, the computation overhead (as well as the accuracy) will be the same as the original model. Finally, Table 16 indicates that MoNE achieves maximum memory reduction that consistently increases with the pruning ratio. For a given pruning ratio, the heavy expert parameters are directly replaced by a light weight constant vector.

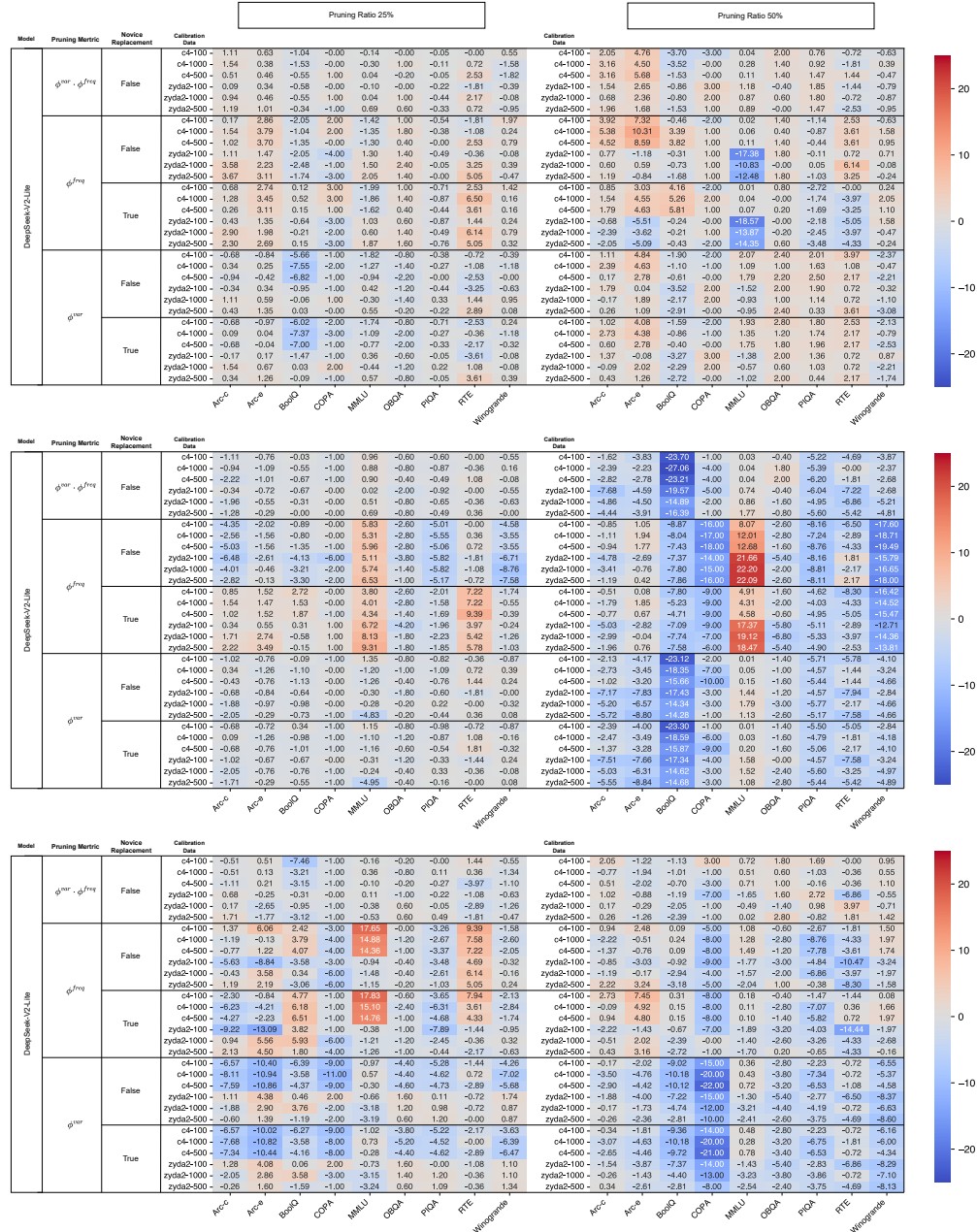

Figure 13: Ablation study on expert access frequency, output variance and novice replacement with detailed results. Numbers are the difference to the proposed MoNE.

