# OpenReview forum: "MoNE: Replacing Redundant Experts with Lightweight Novices for Structured Pruning of MoE"
_ICLR.cc/2026/Conference — ICLR 2026 Poster_

### Official Review · Reviewer_tzmx · 2025-10-31

**Soundness:** 3
**Presentation:** 3
**Contribution:** 3
**Rating:** 6
**Confidence:** 3

**Summary:**

This paper introduces MoNE, a pruning method for Mixture-of-Experts models. It identifies experts with low routing frequency and low output variance, and replaces them with constant vectors called novices. This reduces memory cost while keeping model accuracy stable. Experiments across multiple MoE architectures show that MoNE performs better than existing pruning baselines and remains stable across different calibration settings.

**Strengths:**

The idea of replacing experts with constant novice vectors is simple, easy to implement, and maintains the computational benefits associated with pruning.

The redundancy score combining frequency and output variance is well motivated and avoids relying solely on routing frequency, which previous methods often do.

The experimental evaluation is broad, covering multiple MoE architectures, different pruning ratios, and variations in calibration data, showing consistent robustness.

**Weaknesses:**

Replacing experts with constant vectors may reduce the model's expressive power, and test cases, due to their limitations, may not be able to cover the negative impacts of the evaluation.

Pruning strategies depend on the distribution of the calibration dataset. Different application scenarios may have different dependencies on experts, and differences between the distribution of the calibration dataset and the distribution of the real-world scenario may lead to the erroneous removal of important experts.

**Questions:**

No further questions, see above.

---

> ### Author Response · Authors · 2025-11-21
> **Response to Reviewer tzmx**
>
> > **W1:** Replacing experts with constant vectors may reduce the model's expressive power, and test cases, due to their limitations, may not be able to cover the negative impacts of the evaluation.
> > **W2:** Pruning strategies depend on the distribution of the calibration dataset. Different application scenarios may have different dependencies on experts, and differences between the distribution of the calibration dataset and the distribution of the real-world scenario may lead to the erroneous removal of important experts.
>
> To reveal the model capability on different test cases and show the effectiveness of our pruning strategies on different scenarios, we extended MoNE on two specialized tasks, Math and GSM8K.
> Specifically, we conducted experiments on both base models and instruct models. In addition, we adopted the first 100 samples from the training dataset of the two tasks to calibrate the instruct models. The results are reported in the five tables below and we also updated the details in Appendix A of the revised paper.
> | Model | Calibration dataset | GSM8K | Math |
> |---|:---:|:---:|:---:|
> | OLMoE | - | 52.77 | 15.78 |
> | OLMoE-Instruct | - | 67.63 | 18.64 |
> | OLMoE | Zyda2 | 3.34 | 1.94 |
> | OLMoE-Instruct | Zyda2 | 7.96 | 2.54 |
> | OLMoE-Instruct | Math | 65.28 | 18.22 |
> | OLMoE-Instruct | GSM8K | 67.48 | 17.64 |
>
> | Model | Calibration dataset | GSM8K | Math |
> |---|:---:|:---:|:---:|
> | Moonlight | - | 74.22 | 42.32 |
> | Moonlight-Instruct | - | 77.03 | 39.26 |
> | Moonlight | Zyda2 | 46.40 | 3.32 |
> | Moonlight-Instruct | Zyda2 | 51.86 | 6.64 |
> | Moonlight-Instruct | Math | 75.28 | 38.56 |
> | Moonlight-Instruct | GSM8K | 76.72 | 35.84 |
>
> | Model | Calibration dataset | GSM8K | Math |
> |---|:---:|:---:|:---:|
> | Deepseek-V2-Lite | - | 38.82 | 16.54 |
> | DeepSeek-V2-Lite-Chat | - | 66.49 | 18.56 |
> | Deepseek-V2-Lite | Zyda2 | 25.40 | 6.80 |
> | DeepSeek-V2-Lite-Chat | Zyda2 | 37.07 | 4.76 |
> | DeepSeek-V2-Lite-Chat | Math | 64.44 | 20.76 |
> | DeepSeek-V2-Lite-Chat | GSM8K | 63.76 | 19.70 |
>
> | Model | Calibration dataset | GSM8K | Math |
> |---|:---:|:---:|:---:|
> | Qwen2-57B-A14B | - | 79.68 | 40.50 |
> | Qwen2-57B-A14B-Instruct | - | 69.90 | 31.30 |
> | Qwen2-57B-A14B | Zyda2 | 74.07 | 31.28 |
> | Qwen2-57B-A14B-Instruct | Zyda2 | 64.90 | 22.84 |
> | Qwen2-57B-A14B-Instruct | Math | 58.45 | 29.56 |
> | Qwen2-57B-A14B-Instruct | GSM8K | 52.92 | 29.62 |
>
> | Model | Calibration dataset | GSM8K | Math |
> |---|:---:|:---:|:---:|
> | Qwen3-30B-A3B | - | 85.37 | 50.48 |
> | Qwen3-30B-A3B-Instruct | - | 90.90 | 46.94 |
> | Qwen3-30B-A3B | Zyda2 | 71.72 | 9.44 |
> | Qwen3-30B-A3B-Instruct | Zyda2 | 78.01 | 9.88 |
> | Qwen3-30B-A3B-Instruct | Math | 89.76 | 48.52 |
> | Qwen3-30B-A3B-Instruct | GSM8K | 90.30 | 51.70 |
>
> The conclusions are:
> 1. After pruned by pretraining data (Zyda2), the larger the ***base model*** is, the more accuracy MoNE can preserve, which is consistent with our observations from general tasks in Section 5.2. Yet, **pretraining data indeed cannot accurately capture the distribution of specialized tasks**, leading to up to 49% drop for the smallest model, OLMoE.
> 2. **MoNE can easily enhance the performance via pruning the ***instruct models*** with the training data from the specialized tasks**. For example, OLMoE-Instruc incurs only **<=1%** accurarcy drop for both tasks. This is not too surprising, as SOTA models also rely on specialized finetuning (SFT, RL, etc.) with domain-specific data to enhance their capability in these tasks [1-4].
>
> In short, we acknowledge that **calibration via pretraining data is not sufficient to keep the model capability for specialized tasks, and task-specific calibration can mitigate this issue**. A promising future direction for model compression is to bridge the two points.
>
> [1] Niklas Muennighoff, et al. OLMoe: Open mixture-of-experts language models. In The Thirteenth International Conference on Learning Representations, 2025.
> [2] Jingyuan Liu, et al. Muon is scalable for llm training. arXiv preprint arXiv:2502.16982, 2025.
> [3] Team Qwen. Qwen2 technical report. arXiv preprint arXiv:2407.10671, 2024.
> [4] An Yang, et al. Qwen3 technical report. arXiv preprint arXiv:2505.09388, 2025.

---

### Official Review · Reviewer_cM6H · 2025-10-31

**Soundness:** 3
**Presentation:** 3
**Contribution:** 3
**Rating:** 6
**Confidence:** 5

**Summary:**

The research focus of this paper is expert pruning in Mixture-of-Experts (MoE) models. To address this issue, the paper proposes Mixture-of-Novices-and-Experts (MoNE), a novel expert pruning method that replaces redundant experts with lightweight novices to achieve effective and robust model compression. Experiments demonstrate that MoNE consistently outperforms baseline methods across three dimensions—model architectures, calibration data sources, and calibration sample sizes—with minimal accuracy loss, validating its effectiveness and robustness.

**Strengths:**

1. This paper propose a novel expert pruning method named MoNE which replaces redundant experts with lightweight novices to compress MoE models with minimal performance loss
2. This paper uses expert access frequency and output variance to measure redundancy, and unbiased output estimation to minimize post-pruning discrepancy, yielding effective and robust pruning.

**Weaknesses:**

1. The combinatorial forms of frequency and variance adjacency matrices require ablation, such as weighted summation.
2. Replacing experts with constant vectors may reduce expressiveness; could learnable vectors or biases be used instead?
3. Could comparative experiments on pruning strategies (without finetuning) be provided to demonstrate the superiority of the proposed frequency- and variance-based pruning strategy?

**Questions:**

Refer to Weaknesses

---

> ### Author Response · Authors · 2025-11-21
> **Response to Reviewer cM6H**
>
> > **W1:** The combinatorial forms of frequency and variance adjacency matrices require ablation, such as weighted summation.
>
> We extended ablation study on the redundancy score. Specifically, we follows the same evaluation configuration in Section 5.2 and compare MoNE against the following three variants: weighted sum with 25% / 50% / 75% on output variance. The results are reported in the following table, and we also updated the details in Appendix B of the revised paper.
>
> |  | MoNE | Weighted sum 25% on output variance | Weighted sum 50% on output variance | Weighted sum 75% on output variance |
> |---|---|---|---|---|
> | OLMoE | 61.04 | 57.07 | 57.07 | 57.07 |
> | Moonlight | 69.10 | 68.95 | 69.03 | 69.07 |
> | Deepseek-V2-Lite | 66.12 | 65.63 | 65.66 | 66.07 |
> | Qwen2-57B-A14B | 71.75 | 72.55 | 72.57 | 72.53 |
> | Qwen3-30B-A3B | 72.69 | 71.96 | 72.29 | 72.48 |
>
> The weighted sum results reveal that **only Qwen2-57B-A14B can achieve minor improvement (\~0.8%), but incur severe drop (\~4%) on small models like OLMoE**.
>
> ---
>
> > **W2:** Replacing experts with constant vectors may reduce expressiveness; could learnable vectors or biases be used instead?
>
> Section 5.5 applied continued pretraining to update the model parameters including the novices. In this experiment, novices become learnable. **The results confirm that further training can enhance the model expressiveness, but at the cost of higher compute and data resource.**
>
> Moreover, in training settings, we can **consider MoNE as an effective initialization technique**. As indicated by previous work [1,2], effective structured pruning can largely reduce the compute and data resource to recover the performance loss.
>
> [1] Saurav Muralidharan, et al. Compact language models via pruning and knowledge distillation. Advances in Neural Information Processing Systems, 37:41076–41102, 2024.
> [2] Sharath Turuvekere Sreenivas, et al. Llm pruning and distillation in practice: The minitron approach. arXiv preprint arXiv:2408.11796, 2024.
>
> ---
> > **W3:** Could comparative experiments on pruning strategies (without finetuning) be provided to demonstrate the superiority of the proposed frequency- and variance-based pruning strategy?
>
> To the best of our knowledge, until the submission of this work, the four adopted baselines in Section 5.2 are **the most recent competitive studies with runnable code in the domain of structured model pruning**. The updated ablation study is provided in Section 5.4, and we also provided a pruning metric ablation in Appendix B.
> We would appreciate it if the reviewer could clarify specific comparison target.

---

### Official Review · Reviewer_14Ca · 2025-11-01

**Soundness:** 3
**Presentation:** 3
**Contribution:** 2
**Rating:** 6
**Confidence:** 4

**Summary:**

MoNE proposes to prune MoE models by replacing selected experts with constant novices (per-expert mean outputs) instead of simply deleting or merging them. Redundancy is estimated by two metrics computed on a small calibration set—access frequency and output variance—and the novice for a pruned expert is its unbiased mean output. The method is evaluated across several MoE architectures (OLMoE, Moonlight, DeepSeek-V2-Lite, Qwen2-57B-A14B, Qwen3-30B-A3B), calibration sources/sizes, and pruning ratios, with ablations on the metrics and novice replacement.

**Strengths:**

* Simple, compute-friendly pruning primitive (constant novices) that retains router behavior and keeps overhead close to removal.

* Consistent gains/robustness across models and calibration setups; headline numbers are competitive.

**Weaknesses:**

* The ablation in Figure 4 is intersting, seems like the variance metric can bring improvement without the novice. It would be helpful to include more comprehensive ablation, i.e. more combination (e.g. only frequency and only variance) to show the gain from each part.


* The novice is the unbiased mean output of a pruned expert (a constant vector), similar to FLAP’s use of averaged activations for compensation but at a different granularity. The paper should more explicitly discuss the relation with FALP and isolate the contribution.


* The redundancy score is the product of variance and frequency. This hard-coded fusion may be scale-sensitive; it would be helpful to include normalized scores, log-sum, or a learned weight λ and report stability.


* Some related works worth mentioning [1,2]

[1] MOE-PRUNER: PRUNING MIXTURE-OF-EXPERTS LARGE LANGUAGE MODEL USING THE HINTS FROM ITS ROUTER
[2] SlimMoE: Structured Compression of Large MoE Models via Expert Slimming and Distillation

**Questions:**

* After pruning, do tokens get routed more often to the remaining real experts or to novices? Any trends per layer?

---

> ### Author Response · Authors · 2025-11-21
> **Response to Reviewer 14Ca 1/3**
>
> > **W1:** The ablation in Figure 4 is intersting, seems like the variance metric can bring improvement without the novice. It would be helpful to include more comprehensive ablation, i.e. more combination (e.g. only frequency and only variance) to show the gain from each part.
>
> We have augmented the ablation study with two additional variants: frequency-only without novice replacement and variance-only without novice replacement. The results are presented in the following tables and we also updated the details in Section 5.4 of the revised paper.
> - 25% pruning ratio:
>
> | Pruning metric | Noivce replacement | Arc-c | Arc-e | BoolQ | COPA | MMLU | OBQA | PIQA | RTE | Winogrande |
> |---|---|---|---|---|---|---|---|---|---|---|
> | $\theta^{var}*\theta^{freq}$ | False | -0.11 | -0.28 | -1.39 | -0.39 | +0.19 | -0.22 | -0.29 | -0.16 | -0.63 |
> | $\theta^{freq}$ | False | -1.09 | +0.72 | -1.13 | -2.28 | +4.37 | -0.51 | -2.80 | +2.51 | -2.13 |
> | $\theta^{freq}$ | True | -0.19 | +0.90 | +1.93 | -0.72 | +4.40 | -0.86 | -2.27 | +4.23 | -0.56 |
> | $\theta^{var}$ | False | -1.64 | -1.51 | -2.06 | -2.06 | -0.98 | -1.20 | -0.93 | -0.44 | -0.84 |
> | $\theta^{var}$ | True | -1.57 | -1.45 | -2.08 | -2.11 | -0.97 | -1.21 | -0.88 | -0.58 | -0.84 |
>
> - 50% pruning ratio:
>
> | Pruning metric | Noivce replacement | Arc-c | Arc-e | BoolQ | COPA | MMLU | OBQA | PIQA | RTE | Winogrande |
> |---|---|---|---|---|---|---|---|---|---|---|
> | $\theta^{var}*\theta^{freq}$ | False | -0.45 | -0.43 | -8.07 | -1.33 | +0.37 | +0.67 | -1.21 | -1.87 | -1.23 |
> | $\theta^{freq}$ | False | +0.09 | +1.54 | -2.77 | -7.33 | +3.15 | -1.04 | -4.67 | -1.36 | -5.86 |
> | $\theta^{freq}$ | True | -0.71 | +1.08 | -1.72 | -4.50 | +0.97 | -1.72 | -3.64 | -3.95 | -4.67 |
> | $\theta^{var}$ | False | -1.52 | -2.11 | -8.86 | -6.61 | +0.04 | -1.22 | -2.70 | -1.58 | -4.10 |
> | $\theta^{var}$ | True | -1.43 | -2.11 | -8.85 | -6.33 | +0.05 | -1.13 | -2.70 | -1.68 | -4.04 |
>
> The expanded ablation study yields several key findings:
> 1. **Novice replacement consistently enhances performance:** For both variance-only and frequency-only pruning metrics, incorporating novice replacement leads to performance improvements across most tasks and both pruning ratios. This demonstrates that the novice module serves as an effective substitute for pruned experts.
> 2. **Frequency-only metric exhibits task-specific gains but lacks generalization:** While frequency-based pruning (with or without novice replacement) demonstrates advantages on specific benchmarks such as MMLU and RTE, these gains do not generalize consistently across the complete evaluation suite. This suggests that frequency alone is insufficient as a pruning criterion.
> 3. **The proposed MoNE demonstrates superior overall performance:** Examining the complete ablation table reveals that most alternative configurations yield negative performance deltas relative to our proposed MoNE approach (fused metric with novice replacement). MoNE achieves stronger average performance across both evaluation tasks and pruning ratios, indicating that both components—metric fusion and novice replacement—are necessary and complementary for optimal results.
>
> These findings validate our design choices and underscore the importance of combining the two pruning signals with effective expert replacement strategies.

---

> ### Author Response · Authors · 2025-11-21
> **Response to Reviewer 14Ca 2/3**
>
> > **W2:** The novice is the unbiased mean output of a pruned expert (a constant vector), similar to FLAP’s use of averaged activations for compensation but at a different granularity. The paper should more explicitly discuss the relation with FALP and isolate the contribution.
>
> - Relation
> Both MoNE and FLAP share the idea that pruning certain model components and compensate the performance loss by appending another lightweight component.
>
> - Difference
>
> | Method | Pruning component | Pruning metric | Metric decider | Ranking scope | Compensation |
> |---|---|---|---|---|---|
> | FLAP | Weight channels | Weight channel importance | Input activation channel variance | Weight matrices per layer | averaged output of pruned channels |
> | MoNE | Experts | Expert redundancy | Expert access frequency & output variance | Experts per layer | averaged output of pruned experts |
>
> The differences between MoNE and FLAP are summarized in the above table.
>
> FLAP evaluates the importance of individual weight channels based on the input activation variance of each channel. After removing the less important channels of certain weight matrices, it compensates with a bias term which is the averaged output of the removed channels.
> In contrast, MoNE evaluates the expert redundancy based on the expert access frequency and output variance. After removing redundant experts, it compensates performance loss with novices, the averaged output of pruned experts.
>
> As we indicated in the introduction and verified in Section 5.2, **general structured pruning methods like FLAP are often designed for vanilla Transformer models, and fail to account for the sparse computation scheme of MoE models when evaluating model component importance**, leading to inconsistent performance drop across model architectures, calibration data sources and calibration sample sizes.
>
> ---
>
> > **W3:** The redundancy score is the product of variance and frequency. This hard-coded fusion may be scale-sensitive; it would be helpful to include normalized scores, log-sum, or a learned weight λ and report stability.
>
> We extended ablation study on the redundancy score. Specifically, we follows the same evaluation configuration in Section 5.2 and compare MoNE against the following five variants: normalized scores, log-sum, weighted sum with 25% / 50% / 75% on output variance. The results are reported in the following table, and we also updated the details in Appendix B of the revised paper.
>
> |  | MoNE | Normalized scores | Log-sum | Weighted sum 25% on output variance | Weighted sum 50% on output variance | Weighted sum 75% on output variance |
> |---|:---:|:---:|:---:|:---:|:---:|:---:|
> | OLMoE | 61.04 | 61.39 | 61.23 | 57.07 | 57.07 | 57.07 |
> | Moonlight | 69.1 | 68.58 | 69.05 | 68.95 | 69.03 | 69.07 |
> | Deepseek-V2-Lite | 66.12 | 65.70 | 66.02 | 65.63 | 65.66 | 66.07 |
> | Qwen2-57B-A14B | 71.75 | 71.73 | 71.75 | 72.55 | 72.57 | 72.53 |
> | Qwen3-30B-A3B | 72.69 | 73.62 | 73.66 | 71.96 | 72.29 | 72.48 |
>
> **The results indicate that the pruning score is not scale-sensitive.**
> In particular, log-sum shows almost identical results to MoNE, as $\log(\theta^{var}) + \log(\theta^{freq}) = \log(\theta^{var}*\theta^{freq})$ does not affect the partial order during expert redundancy ranking.
> We do not include learnable weight, as training five models are too expensive and time-consuming. However, the weighted sum results reveal that only Qwen2-57B-A14B can achieve minor improvement (\~0.8%), but incur severe drop (\~4%) on small models like OLMoE.
>
> ---
>
> > **W4:** Some related works worth mentioning [1,2]
>
> MOE-PRUNER [1] prunes weights with the smallest magnitudes multiplied by the corresponding input activations and router weights. SlimMoE [2] use the top-8 routing scores distillation loss to compute the sensitivity score for  each parameter.
> Both methods **ignore the expert output variance when searching the pruning targets**. Moreover, MOE-PRUNER does not evaluate the **robustness in model architectures and calibration datasets**, and SlimMoE **needs continual pre-training with up to 400B tokens**.
> We updated the discussion in Section 2 of the revised paper.
>
> [1] MOE-PRUNER: PRUNING MIXTURE-OF-EXPERTS LARGE LANGUAGE MODEL USING THE HINTS FROM ITS ROUTER
> [2] SlimMoE: Structured Compression of Large MoE Models via Expert

---

> ### Author Response · Authors · 2025-11-21
> **Response to Reviewer 14Ca 3/3**
>
> > **Q1:** After pruning, do tokens get routed more often to the remaining real experts or to novices? Any trends per layer?
>
> We have counted the tokens routed to each expert/novice in each task for the five adopted models, both original and pruned version. Then we have calculated ratio of token routing change for novices in each layer. The statistics are summarized in the following five tables.
> The results indicate that **MoNE has negligible affect (\~1% on average) on the token routing**.
>
> - OLMoE
>
> |  | Arc-c & Arc-e | BoolQ | COPA | MMLU | OBQA | PIQA | RTE | Winogrande |
> |---|:---:|:---:|:---:|:---:|:---:|:---:|---|---|
> | Total tokens per expert/novice | 415056 | 495616 | 3472 | 1753216 | 22648 | 130328 | 26776 | 54096 |
> | min - max (%) | -3.11 - 6.71 | -1.43 - 4.02 | -3.08 - 2.39 | -2.02 - 3.03 | -5.43 - 1.62 | -2.26 - 1.68 | -3.02 - 0.99 | -4.40 - 1.22 |
> | avg (std) (%) | 0.05 (2.10) | -0.27 (1.28) | -0.02 (1.57) | 0.27 (1.10) | -0.46 (1.69) | -0.14 (1.00) | -0.94 (1.15) | -0.63 (1.42) |
>
> - Moonlight
>
> |  | Arc-c & Arc-e | BoolQ | COPA | MMLU | OBQA | PIQA | RTE | Winogrande |
> |---|:---:|:---:|:---:|:---:|:---:|:---:|---|---|
> | Total tokens per expert/novice | 304110 | 367800 | 2472 | 1271520 | 16248 | 94434 | 19152 | 39564 |
> | min - max (%) | -0.08 - 2.51 | -0.33 - 0.40 | -1.09 - 4.29 | -0.40 - 2.62 | -0.40 - 2.63 | -0.42 - 1.35 | -2.63 - 3.35 | -0.31 - 3.16 |
> | avg (std) (%) | 1.12 (0.79) | 0.11 (0.14) | 0.57 (1.10) | 1.05 (0.83) | 0.98 (0.80) | 0.53 (0.38) | 0.65 (1.11) | 0.72 (0.85) |
>
> - DeepSeek-V2-Lite
>
> |  | Arc-c & Arc-e | BoolQ | COPA | MMLU | OBQA | PIQA | RTE | Winogrande |
> |---|:---:|:---:|:---:|:---:|:---:|:---:|---|---|
> | Total tokens per expert/novice | 311592 | 388266 | 2412 | 1346904 | 16362 | 97410 | 19926 | 39756 |
> | min - max (%) | -0.33 - 0.71 | -0.16 - 1.06 | -0.91 - 1.33 | -0.28 - 1.45 | -0.17 - 0.68 | -0.67 - 1.32 | -0.63 - 2.02 | -0.23 - 0.70 |
> | avg (std) (%) | 0.21 (0.27) | 0.34 (0.32) | 0.15 (0.61) | 0.46 (0.36) | 0.19 (0.22) | 0.32 (0.44) | 0.40 (0.63) | 0.16 (0.25) |
>
> - Qwen2-57B-A14B
>
> |  | Arc-c & Arc-e | BoolQ | COPA | MMLU | OBQA | PIQA | RTE | Winogrande |
> |---|:---:|:---:|:---:|:---:|:---:|:---:|---|---|
> | Total tokens per expert/novice | 408192 | 504096 | 3280 | 1717632 | 22208 | 128024 | 25664 | 52880 |
> | min - max (%) | -1.26 - 0.50 | -1.49 - 0.28 | -1.22 - 1.59 | -1.33 - 0.51 | -0.83 - 0.97 | -1.20 - 0.34 | -3.75 - 1.75 | -0.59 - 0.48 |
> | avg (std) (%) | -0.16 (0.43) | -0.21 (0.38) | -0.00 (0.56) | -0.10 (0.38) | 0.06 (0.40) | -0.23 (0.44) | -0.37 (1.03) | -0.09 (0.29) |
>
> - Qwen3-30B-A3B
>
> |  | Arc-c & Arc-e | BoolQ | COPA | MMLU | OBQA | PIQA | RTE | Winogrande |
> |---|:---:|:---:|:---:|:---:|:---:|:---:|---|---|
> | Total tokens per expert/novice | 408192 | 504096 | 3280 | 1717632 | 22208 | 128024 | 25664 | 52880 |
> | min - max (%) | -1.37 - 2.67 | -0.13 - 0.39 | -1.22 - 0.73 | -1.10 - 1.88 | -0.56 - 0.88 | -1.10 - 2.57 | -1.97 - 3.23 | -0.47 - 0.74 |
> | avg (std) (%) | 0.66 (0.80) | 0.11 (0.11) | 0.15 (0.38) | 0.51 (0.59) | 0.24 (0.29) | 0.81 (0.74) | 0.41 (0.84) | 0.15 (0.22) |

---

### Official Review · Reviewer_qHPd · 2025-11-03

**Soundness:** 3
**Presentation:** 3
**Contribution:** 3
**Rating:** 6
**Confidence:** 2

**Summary:**

This paper proposes MoNE, a structured pruning approach that replaces redundant experts with lightweight "novices" - essentially constant vectors representing averaged expert outputs. The key idea is identifying redundant experts using two metrics: access frequency (how often an expert is selected) and output variance (how stable an expert's outputs are). Experts with low frequency and low variance get replaced by their mean output. The authors test this on five MoE models (OLMoE, Moonlight, DeepSeek-V2-Lite, Qwen2-57B-A14B, Qwen3-30B-A3B) at 25% and 50% pruning ratios, showing better performance than existing methods like MC-SMoE, RS, Angular, and FLAP.

**Strengths:**

1. The core idea is intuitive, simple, and training-free. The fused metric (frequency + variance) is well-justified, and the "novice" replacement (the expert's mean output) is an effective closed-form solution to minimize output discrepancy.

2. The experimental validation is a major strength. Testing on five different MoE architectures with varying sizes (7B to 57B parameters) demonstrates the method works across scales. The robustness evaluation across model architectures, calibration data sources (Zyda2 vs C4), and sample sizes (100, 500, 1000) is thorough.

**Weaknesses:**

1. There is a lack of specialized tasks (e.g., coding, math) in evaluation. It's unclear if the redundancy metric, calibrated on general text, might inadvertently prune experts that are critical for these specialized capabilities.

2. The paper doesn't explain or ablate the benefit of computing a dynamic, per-token gate for a static, constant "novice" vector. This appears computationally redundant.

**Questions:**

1. How does MoNE perform on specialized benchmarks like Math or GSM8K? The current evaluation on general tasks may not be sufficient to prove that specialized experts are not being harmed.

2. What is the justification for per-token routing to a static novice vector? Would a simpler approach, like adding the novice as a scaled, static bias, perform comparably while saving routing computation?

3. Are novices trainable during continued pretraining?

---

> ### Author Response · Authors · 2025-11-21
> **Response to Reviewer qHPd 1/2**
>
> > **W1/Q1:** How does MoNE perform on specialized benchmarks like Math or GSM8K? The current evaluation on general tasks may not be sufficient to prove that specialized experts are not being harmed.
>
> We extended MoNE on two specialized tasks, Math and GSM8K. We conducted experiments on both base models and instruct models. In addition, we adopted the first 100 samples from the training dataset of the two tasks to calibrate the instruct models. The results are reported in the five tables below and we also updated the details in Appendix A of the revised paper.
> | Model | Calibration dataset | GSM8K | Math |
> |---|:---:|:---:|:---:|
> | OLMoE | - | 52.77 | 15.78 |
> | OLMoE-Instruct | - | 67.63 | 18.64 |
> | OLMoE | Zyda2 | 3.34 | 1.94 |
> | OLMoE-Instruct | Zyda2 | 7.96 | 2.54 |
> | OLMoE-Instruct | Math | 65.28 | 18.22 |
> | OLMoE-Instruct | GSM8K | 67.48 | 17.64 |
>
> | Model | Calibration dataset | GSM8K | Math |
> |---|:---:|:---:|:---:|
> | Moonlight | - | 74.22 | 42.32 |
> | Moonlight-Instruct | - | 77.03 | 39.26 |
> | Moonlight | Zyda2 | 46.40 | 3.32 |
> | Moonlight-Instruct | Zyda2 | 51.86 | 6.64 |
> | Moonlight-Instruct | Math | 75.28 | 38.56 |
> | Moonlight-Instruct | GSM8K | 76.72 | 35.84 |
>
> | Model | Calibration dataset | GSM8K | Math |
> |---|:---:|:---:|:---:|
> | Deepseek-V2-Lite | - | 38.82 | 16.54 |
> | DeepSeek-V2-Lite-Chat | - | 66.49 | 18.56 |
> | Deepseek-V2-Lite | Zyda2 | 25.40 | 6.80 |
> | DeepSeek-V2-Lite-Chat | Zyda2 | 37.07 | 4.76 |
> | DeepSeek-V2-Lite-Chat | Math | 64.44 | 20.76 |
> | DeepSeek-V2-Lite-Chat | GSM8K | 63.76 | 19.70 |
>
> | Model | Calibration dataset | GSM8K | Math |
> |---|:---:|:---:|:---:|
> | Qwen2-57B-A14B | - | 79.68 | 40.50 |
> | Qwen2-57B-A14B-Instruct | - | 69.90 | 31.30 |
> | Qwen2-57B-A14B | Zyda2 | 74.07 | 31.28 |
> | Qwen2-57B-A14B-Instruct | Zyda2 | 64.90 | 22.84 |
> | Qwen2-57B-A14B-Instruct | Math | 58.45 | 29.56 |
> | Qwen2-57B-A14B-Instruct | GSM8K | 52.92 | 29.62 |
>
> | Model | Calibration dataset | GSM8K | Math |
> |---|:---:|:---:|:---:|
> | Qwen3-30B-A3B | - | 85.37 | 50.48 |
> | Qwen3-30B-A3B-Instruct | - | 90.90 | 46.94 |
> | Qwen3-30B-A3B | Zyda2 | 71.72 | 9.44 |
> | Qwen3-30B-A3B-Instruct | Zyda2 | 78.01 | 9.88 |
> | Qwen3-30B-A3B-Instruct | Math | 89.76 | 48.52 |
> | Qwen3-30B-A3B-Instruct | GSM8K | 90.30 | 51.70 |
>
> The conclusions are:
> 1. After pruned by pretraining data (Zyda2), the larger the ***base model*** is, the more accuracy MoNE can preserve, which is consistent with our observations from general tasks in Section 5.2. Yet, pretraining data indeed cannot accurately capture the distribution of specialized tasks, leading to up to 49% drop for the smallest model, OLMoE.
> 2. MoNE can easily enhance the performance via pruning the ***instruct models*** with the training data from the specialized tasks. For example, OLMoE-Instruc incurs only **<=1%** accurarcy drop for both tasks. This is not too surprising, as SOTA models also rely on specialized finetuning (SFT, RL, etc.) with domain-specific data to enhance their capability in these tasks [1-4].
>
> In short, **we acknowledge that calibration via pretraining data is not sufficient to keep the model capability for specialized tasks, and task-specific calibration can resolve this issue**. A promising future direction for model compression is to bridge the two points.
>
> [1] Niklas Muennighoff, et al. OLMoe: Open mixture-of-experts language models. In The Thirteenth International Conference on Learning Representations, 2025.
> [2] Jingyuan Liu, et al. Muon is scalable for llm training. arXiv preprint arXiv:2502.16982, 2025.
> [3] Team Qwen. Qwen2 technical report. arXiv preprint arXiv:2407.10671, 2024.
> [4] An Yang, et al. Qwen3 technical report. arXiv preprint arXiv:2505.09388, 2025.

---

> ### Author Response · Authors · 2025-11-21
> **Response to Reviewer qHPd 2/2**
>
> > **W2/Q2:** What is the justification for per-token routing to a static novice vector? Would a simpler approach, like adding the novice as a scaled, static bias, perform comparably while saving routing computation?
>
> First, we would like to clarify that each pruned expert is replaced with a separate novice after calibration, instead of a single novice. During inference, tokens routed to the original expert will skip the computation and the output is changed to the corresponding novice.
>
> Second, the representation of novice vector can be proved to be the best statistical approximation of the original expert output:
> To minimize the performance loss after replacing expert $E_i$ with novice $N_i$ over a calibration dataset $\mathcal{C}$, the problem is formulated as
> $$
> \min\sum_{x \in \mathcal{C}}(\Vert E_i(x) -N_i(x) \Vert_2)
> $$
> Let $N_i$ be the target, this equation can be resolved by $N_i=\overline{E_i}$, which is the averaged output of $E_i$.
> Next, let $E(x)$ be the target, we can get that
> $$
> \mathbb{E}(\Vert E_i(x) -N_i(x) \Vert_2) = \mathbb{E}(\Vert E_i(x) -\overline{E_i} \Vert_2) = Var(E_i(x))
> $$
> Thus, low-variance experts intrinsically incur low approximation error when replaced by novices.
>
> ---
>
> >  **Q3:**  Are novices trainable during continued pretraining?
>
> Yes. Continued pretraining updates all parameters including novices, experts and attention modules.

---

### Author Response · Authors · 2025-11-22
**Overall response to all reviewers**

We thank the reviewers for all the constructive feedbacks and suggestions. We are happy that the reviewers commonly have a lot of positive feedback for the work, including:

- "The idea is intuitive, simple, compute-friendly and effective." (Reviewer qHPd, 14Ca, tzmx)
- "The experiments are comprehensive and results are competitive." (Reviewer qHPd, 14Ca, cM6H, tzmx)

We have carefully addressed the reviewer comments accordingly, and summarized the responses with augmented experiments in the following.

1. Generalization to specialized tasks
To address the concerns of Reviewer qHPd and tzmx, we applied MoNE to specialized tasks. experiment results show that pretraining data indeed cannot accurately capture the distribution of specialized tasks, leading to significant performance drop for models pruned by pretraining data. However, MoNE can easily enhance the performance via pruning the instruct models with the training data from the specialized tasks, resulting in **performance drop within 1%**. We have updated the details in Appendix A of the revised paper.

2. More ablation study on the impact on the pruning metrics and novice
To address the concern of Reviewer 14Ca, we appended two variants: frequency-only pruning without novice replacement, variance-only pruning without novice. The results indicate that novice replacement consistently enhances performance and frequency-only metric exhibits task-specific gains but lacks generalization. Overall, **MoNE demonstrates superior and consistent performance perserving**. We have updated the details in Section 5.4 of the revised paper.

3. Ablation study on the pruning metric variants
To address the concerns of Reviewer 14Ca and cM6H, we tested various combination of frequency and variance signals. The results indicate that the product of variance and frequency is **not scale-sensitive** (Reviewer 14Ca), and weighted sum can achieve only minor improvement (\~0.8%) for Qwen2-57B-A14B, but **incur severe drop (\~4%) for smaller models** (Reviewer cM6H).
We have updated the details in Appendix B of the revised paper.

4. MoE routing invariance after pruning
To address the concern of Reviewer 14Ca, we calculated the ratio of token routing change for novices in each layer. The statistics indicate that **MoNE has negligible affect (\~1% on average) on the token routing**.

---

### Meta-Review · Area_Chair_mNve · 2026-01-13

**Summary:**

The core idea of replacing redundant experts with lightweight novices rather than simply pruning them is both simple and practical, and I appreciate that it comes with a clean mathematical justification for why novices serve as optimal approximations for low-variance experts. The method demonstrates strong robustness across five different MoE architectures and various calibration settings, which gives me confidence in its practical utility. I was initially concerned about performance on specialized tasks like Math and GSM8K, but the authors' additional experiments showing that task-specific calibration keeps accuracy drops under 1% for instruct models addressed this effectively. While the method does require appropriate calibration data for specialized domains, I consider this a reasonable tradeoff for a training-free approach to MoE memory overhead. As a result, I recommend acceptance for this submission.

**Reviewer Concerns:**

The core idea this work is simple and practical: rather than just dropping experts, replace them with cheap constant approximations. It works across five different MoE architectures and holds up under different calibration settings, which is exactly the kind of robustness you want to see. The reviewers raised fair concerns about specialized tasks like Math and GSM8K, and the authors responded with thorough experiments showing that task-specific calibration keeps accuracy drops under 1% for instruct models. The ablations now properly isolate the contribution of each component, and there is a clean mathematical justification for why novices are optimal approximations for low-variance experts. One limitation worth noting: the method still needs appropriate calibration data for specialized domains, but this is a reasonable tradeoff for a training-free approach. Overall, this is solid work that offers a practical solution to MoE memory overhead.

**Reviewer Scores:**

All four reviewers scored the paper at 6, indicating borderline positive assessments. The authors addressed every technical concern with new experiments and analysis. The consensus supports acceptance.

---

### Decision · Program_Chairs · 2026-01-26

Accept (Poster)